# Convergent evolution and horizontal gene transfer in Arctic Ocean microalgae

Richard G Dorrell[1,2], Alan Kuo[3], Zoltan Füssy[4], Elisabeth H Richardson[5], Asaf Salamov[3], Nikola Zarevski[1,2], Nastasia J Freyria[6], Federico M Ibarbalz[1,2], Jerry Jenkins[3,7], Juan Jose Pierella Karlusich[1,2], Andrei Stecca Steindorff[3], Robyn E Edgar[6], Lori Handley[7], Kathleen Lail[3], Anna Lipzen[3], Vincent Lombard[10], John McFarlane[5], Charlotte Nef[1,2], Anna MG Novák Vanclová[1,2], Yi Peng[3], Chris Plott[7], Marianne Potvin[6], Fabio Rocha Jimenez Vieira[1,2], Kerrie Barry[3], Colomban de Vargas[2,9], Bernard Henrissat[8,10], Eric Pelletier[2,11], Jeremy Schmutz[3,7], Patrick Wincker[2,11], Joel B Dacks[5], Chris Bowler[1,2], Igor V Grigoriev[3,12], Connie Lovejoy[6]

Microbial communities in the world ocean are affected strongly by oceanic circulation, creating characteristic marine biomes. The high connectivity of most of the ocean makes it difficult to disentangle selective retention of colonizing genotypes (with traits suited to biome specific conditions) from evolutionary selection, which would act on founder genotypes over time. The Arctic Ocean is exceptional with limited exchange with other oceans and ice covered since the last ice age. To test whether Arctic microalgal lineages evolved apart from algae in the global ocean, we sequenced four lineages of microalgae isolated from Arctic waters and sea ice. Here we show convergent evolution and highlight geographically limited HGT as an ecological adaptive force in the form of PFAM complements and horizontal acquisition of key adaptive genes. Notably, ice-binding proteins were acquired and horizontally transferred among Arctic strains. A comparison with *Tara* Oceans metagenomes and metatranscriptomes confirmed mostly Arctic distributions of these IBPs. The phylogeny of Arctic-specific genes indicated that these events were independent of bacterial-sourced HGTs in Antarctic Southern Ocean microalgae.

## Introduction

Four marine provinces or biomes have been defined based on temperature, salinity, and upper ocean mixing regimes (1, 2). Three of these, polar, westerlies, and trade wind biomes, correlate with latitude, with the fourth being the coastal boundary zone. Recent global surveys of marine single-celled plankton have revealed that microbial communities are distinct within the biogeographic provinces, with the polar regions furthermore separating into Arctic and Antarctic biomes or zones (3, 4). Although superficially similar environments, with extreme seasonality and ice cover much of the year, there are characteristics of the Arctic that are favorable for the emergence of a distinct Arctic microbiome. The Arctic Ocean is geographically isolated, and the surface waters are fresher because of inflow from large rivers (5) and the transfer of atmospheric moisture from south to north (6). It is a semi-enclosed ocean lying in a basin between the two largest landmasses in the world, North America and Eurasia. In contrast, the Antarctic Southern Ocean surrounds the ice-covered landmass of Antarctica and borders the Atlantic, Pacific, and Indian Oceans. The Arctic is relatively old; an ocean has been present at the pole since the beginning of the Cretaceous. Shaped by tectonic processes (7), the Arctic Ocean has been a relatively closed basin since the Maastrichtian at the end of

[1]Institut de Biologie de l'ENS, Département de Biologie, École Normale Supérieure, CNRS, INSERM, Université PSL, Paris, France [2]CNRS Research Federation for the Study of Global Ocean Systems Ecology and Evolution, FR2022/Tara Oceans GOSEE, Paris, France [3]US Department of Energy Joint Genome Institute, Lawrence Berkeley National Laboratory, Berkeley, CA, USA [4]Department of Parasitology, BIOCEV, Faculty of Science, Charles University, Prague, Czech Republic [5]Division of Infectious Diseases, Department of Medicine, University of Alberta and Department of Biological Sciences, and University of Alberta, Edmonton, Canada [6]Département de Biologie, Institut de Biologie Intégrative des Systèmes, Université Laval, Quebec, Canada [7]HudsonAlpha Institute for Biotechnology, Huntsville, AL, USA [8]Architecture et Fonction des Macromolécules Biologiques, CNRS, Aix-Marseille Université, Marseille, France [9]Sorbonne Université, CNRS, Station Biologique de Roscoff, AD2M, UMR 7144, Roscoff, France [10]Department of Biological Sciences, King Abdulaziz University, Jeddah, Saudi Arabia [11]Génomique Métabolique, Genoscope, Institut de Biologie François Jacob, Commissariat à l'Énergie Atomique, CNRS, Université Évry, Université Paris-Saclay, Évry, France [12]Department of Plant and Microbial Biology, University of California Berkeley, Berkeley, CA, USA

Correspondence: Connie.Lovejoy@bio.ulaval.ca
Elisabeth H Richardson's present address is Mount Royal University, Calgary, Canada
Federico M Ibarbalz's present address is Centro de Investigaciones del Mar y la Atmósfera, CONICET, Universidad de Buenos Aires, Buenos Aires, Argentina
Juan Jose Pierella Karlusich's present address is FAS, Division of Science, Harvard University, Cambridge, MA, USA

 

the late Cretaceous epoch (ca. 70 million years ago [mya]), with episodic sea ice cover since that time (8). This long history suggests possible limited gene flow into the Arctic from the global ocean over potentially vast time scales. The semi-isolation of the Arctic has led to unique ecosystems and is directly linked to the evolution of numerous high-latitude marine mammals (9). In addition, smaller species such as polar cod and ice-dependent Crustacea are found only in the Arctic (10).

Nonetheless, the Arctic has not been static, and over time, gateways, which are narrow regions where water from other oceans can enter and exit the Arctic, have opened and closed. In addition, there have been glacial controls on sea level and swings in the paleoclimate, providing opportunities for temperate and boreal species to colonize the Arctic Ocean (11). During glacial epochs, physical barriers ensure isolation of the sub-Arctic and Arctic populations, allowing genetic divergence (12). These processes would suggest Arctic microbes, including microalgae, have varied biogeographic histories with present conditions selecting coherent metacommunities (13). In addition, the water from five major rivers and numerous smaller rivers that drain vast expanses of the North American and Eurasian continents (2) could be an additional source of species and genetic diversity of microalgae in Arctic seas (14).

The combination of multi-year sea ice and seasonal ice cover defines the Arctic Ocean and surrounding seas as an ice-influenced ecosystem. Cryotolerant and cryophilic photosynthetic eukaryotic microalgae inhabiting both the water column (phytoplankton) and sea ice (sea ice algae) support a marine Arctic food web and Indigenous communities that are inextricably linked to ice (15). The Arctic microalgae proliferate within and under sea ice and in ice-free summer waters near 0°C (16, 17) and persist through the long months of the polar night (18). Despite most of the global biosphere being below 5°C (19), relatively few cold-adapted species have been genetically investigated (20). A handful of polar algal genomes have been extensively studied, with four of these from around Antarctica and classified as psychrophiles, which is commonly defined as not being able to grow above 15°C (19). Specifically, research has been carried out on the sea ice–associated diatom *Fragilariopsis cylindrus* (21) and two green algae that are closely related to each other, *Chlamydomonas* sp. ICE-L from sea ice and sp. UWO241 that was formally known as *C. raudensis* and now *Chlamydomonas priscuii* (22) originally isolated from a meromictic ice-covered dry valley lake (23) and the bi-polar dinoflagellate *Polarella glacialis*. For the Arctic, only an Arctic *P. glacialis* strain has been comparatively well studied (24), to date.

The potential for lineages of ancient Arctic origin and the episodic input of outside species led us to our hypothesis that Arctic microalgae convergently evolved traits or adaptations aiding survival in an ice-influenced ocean. The alternative hypotheses would be that adaptive metabolic pathways and genes were ancient and vertically inherited. A third possibility is that useful genes are acquired from other organisms in the environment, which is the case for bacterial origin ice-binding proteins (IBPs) in Antarctic diatoms (25). To address these questions, we sequenced four microalgae from four classes belonging to three algal phyla: Cryptophyceae (Cryptophyta), Pavlovophyceae (Haptophyta), Chrysophyceae, and Pelagophyceae (both in the Ochrophyta). The three phyla separated during the Mesoproterozoic ~1,000 mya, with

the two ochrophyte lineages separating before the Cambrian epoch 540 mya (26). All four strains for sequencing were originally collected in 1998 from the North Water region of Pikialasorsuaq (Northern Baffin Bay) north of 77°N, where ice flow persists through June (27). The waters are classified as Arctic origin (28), and south flowing currents eventually enter the North Atlantic via the Labrador Current (2, 29). The four isolates were selected from cryopreserved strains that had been deposited in the Bigelow National Center for Algae. The cryptomonad, *Baffinella* sp. CCMP2293, and haptophyte, *Pavlovales* sp. CCMP2436, were isolated from the upper water column, whereas the chrysophyte *Ochromonas* sp. CCMP2298 was isolated from the bottom of a sea ice core (30) and the pelagophyte CCMP2097 from a brine pocket on the sea ice surface (31).

To test our hypothesis, we compared the four newly sequenced genomes to 17 algal genomes from the Joint Genome Institute (JGI) and elsewhere (32) and with transcriptomes from 296 curated strains from Marine Microbial Eukaryote Transcriptome Sequencing Project (MMETSP) (33), creating a pan-algal dataset of potential genes of interest. Alongside the non-Arctic and Southern Ocean strains, the MMETSP dataset includes transcripts from the same four strains and 12 other Arctic marine strains, some of which were isolated during the same 1998 study as the four newly sequenced strains. We searched this pan-algal dataset for putatively Arctic-associated PFAMs and once identified, carried out PFAM gene-specific searches across publicly available datasets to construct more detailed phylogenies of Arctic-enriched gene families. The specificity and distribution of candidate genes in the Arctic Ocean were tested by explicitly mapping homologues retrieved from the metagenomes and metatranscriptomes in the large global *Tara* Oceans expedition dataset.

Our results revealed a quantifiable convergence in genome expansions and PFAM domain content across taxonomically diverse Arctic microalgae, both of which were independent of underlying phylogenetic affiliation. In addition, phylogenies and geographic distributions of target gene families were consistent with inter-algal horizontal gene transfer (HGT) in the Arctic marine environment. The most remarkable example was seen in the diversity and prevalence of ice-binding proteins (IBPs, PFAM—PF11199) among Arctic strains, where we detected at least four distinct clades of IBP genes. The clades were independent from bacterial- and fungal-associated HGT transfers in microalgal strains from coastal Antarctica and the Southern Ocean. The multiple genetic modifications in Arctic microalgae reveal biogeographically constrained trait selectivity as an adaptive force in algal evolution.

# Results

### New genome sequences and phylogenies of four Arctic strains

Assembly length of the draft quality genomes of the four strains ranged from 61.1 Mbp for the *Ochromonas* sp. CCMP2298 to 535 Mbp for the cryptomonad *Baffinella* sp. CCMP2293. Scaffold L50 ranged from 7.5 Kbp for *Ochromonas* sp. CCMP3298 to 439.3 Kbp for *Baffinella*. BUSCO coverage ranged from 72.3% (*Baffinella* CCMP2293) to 89.1% (the

**Table 1. Genome quality of the four Arctic strains.**

| Strain | CCMP2293 | CCMP2436 | CCMP2298 | CCMP2097 |
|---|---|---|---|---|
| Algal class | Cryptophyceae | Pavlovophyceae | Chrysophyceae | Pelagophyceae |
| Assembly length (Mbp) | 534.5 | 165.4 | 61.1 | 85.8 |
| Repeat space (Mbp) | 219.5 | 10.2 | 9.55 | 23.2 |
| Repeat space (%) | 41.1 | 10.2 | 15.6 | 27.0 |
| Scaffolds | 10,880 | 2,243 | 12,764 | 1716 |
| Scaffold N50 | 344 | 196 | 2096 | 99 |
| Scaffold L50, Kbp | 439.3 | 252.3 | 7.5 | 186.1 |
| GC (%) | 53.9 | 55.2 | 53.4 | 62 |
| Haploid gene models | 33,051 | 26,034 | 20,195 | 19,402 |
| Busco coverage (%) | 72.3 | 81.5 | 80.2 | 89.1 |

% BUSCO coverage (single copy, duplicated, fragmented eukaryotic models).

pelagophyte CCMP2097) (Table 1). We first extracted single-copy marker nuclear genes from the genomes to verify the phylogenetic placement of the new genomes. The concatenated homologues of 250 single-copy marker nuclear genes (39,504 aa), from 391 eukaryotic genomes and transcriptomes, included additional cultured Arctic lineages (17). The multigene tree topology of this pan-algal dataset was consistent with other recent phylogenies (26, 34) (Fig 1 and Supplemental Data 1, sheets 1–3). To refine the taxonomic placements, we mined the NCBI nr database for related 18S rRNA (nuclear) and 16S rRNA (chloroplast) gene sequences (referred to as rDNA) for each of the four new genomes (Supplemental Data 1, sheets 4–7) and combined these with verified 18S rRNA sequences in the MMETSP dataset. At the lowest taxonomic level, except for the cryptomonad (*Baffinella*, Fig S1), the isolates are taxonomically distinct from any formally described species and therefore remain to be described. The densely sampled single-gene tree of 18S rDNA placed *Baffinella* sp. CCMP2293 as a conspecific of *Baffinella frigidus* CCMP2045 (Fig S1), which was also isolated from the North Water, as reported previously (35). For both the 18S and 16S gene trees, the novel pelagophyte CCMP2097 was closest to an *Ankylochrysis* (36) clade but still distant from *A. lutea* and several strains isolated from the Beaufort Sea (RCC strains in Fig S2) that were clearly within the *Ankylochrysis* clade (Fig S2). Our haptophyte, *Pavlovales* CCMP2436 (Fig S3), was at the base of a clade within the Pavlovophyceae (Fig S3). The *Ochromonas* sp. CCMP2298 appeared as a sister taxon to a temperate marine isolate *Ochromonas* sp. CCMP1393 (Figs 1 and S4), as previously reported (37). We note that the genus *Ochromonas* is polyphyletic, but our Arctic isolate was in the same clade as the type species for the genus *Ochromonas*, *Ochromonas triangulata*, so is likely a true *Ochromonas* species (38).

Next, we inferred the present-day distributions of the four strains, comparing 18S rDNA (V4 and V9 variable regions) and 16S rDNA (V4-V5 regions), assigned to each strain via stringent criteria (phylogenetic reconciliation + 99% BLASTn similarity threshold) to curated ribotype sequences available from the *Tara* Oceans expedition, which includes the Arctic Polar Circle data (39), (Supplemental Data 1, sheets 8–11). The four strains were detected most frequently in nanoplankton (here defined as the 3–20 µm size fraction) and non-size–fractionated surface water samples (Fig S5).

*Baffinella* sp. CCMP2293 and the pelagophyte CCMP2097 were relatively abundant, with over 50,000 and 10,000 total mapped ribotypes, respectively. Both ribotypes were sporadically detected in coastal boundary zone stations outside the Arctic, but only in the non-size–fractionated data, which may suggest that closely related species are associated with particulates in coastal zones (Fig S6). *Pavlovales* sp. CCMP2436 was relatively rare (60 mapped ribotypes total) but exclusively found in the Arctic. *Ochromonas* sp. CCMP2298 was only detected in the 16S chloroplast V4-V5 search (52 mapped ribotypes) but was likewise Arctic-exclusive (Figs S5 and S6). Overall, the biogeographic results are consistent with a pan-Arctic distribution of the strains and indicate that the distribution of the four strains was overwhelmingly Arctic (Fig S6).

## Arctic algae possess expanded genomes with distinct composition

The genomes of the four Arctic-sourced strains were typically larger than non-Arctic genomes from the sequenced nearest related lineages (NRL) of each Arctic strain. This was most striking for the *Baffinella* sp. CCMP2293 genome (534.5 Mbp, Table 1) that was 6.1 times that of the non-Arctic cryptomonad *Guillardia theta* (87.2 Mbp). More moderate expansions, between 1.2 and 2.1 times the size of the genomes of NRL, were found for other Arctic algae (Supplemental Data 2, sheet 1).

The combined genome and transcriptome pan-algal dataset included 15 Arctic strains and 21 Antarctic strains isolated from south of the Antarctic Polar Front where waters are less than 6°C in summer (40, 41). The remaining pan-algal set consisted of mostly marine isolates from temperate latitudes (Fig S7 and Supplemental Data 2, sheet 2) along with a few estuarine, brackish, and freshwater strains from related lineages. The protein coding potential (genes) of all individual genomes and transcriptomes was extracted for each genome and transcriptome in the pan-algal dataset. We then compared the mean number of genes to known PFAM domains for Arctic and non-polar strains. For the four new genomes, the mean Arctic gene count to known PFAMs was 1.97 compared with the non-polar strains with a mean gene count to known PFAM ratio of 1.72

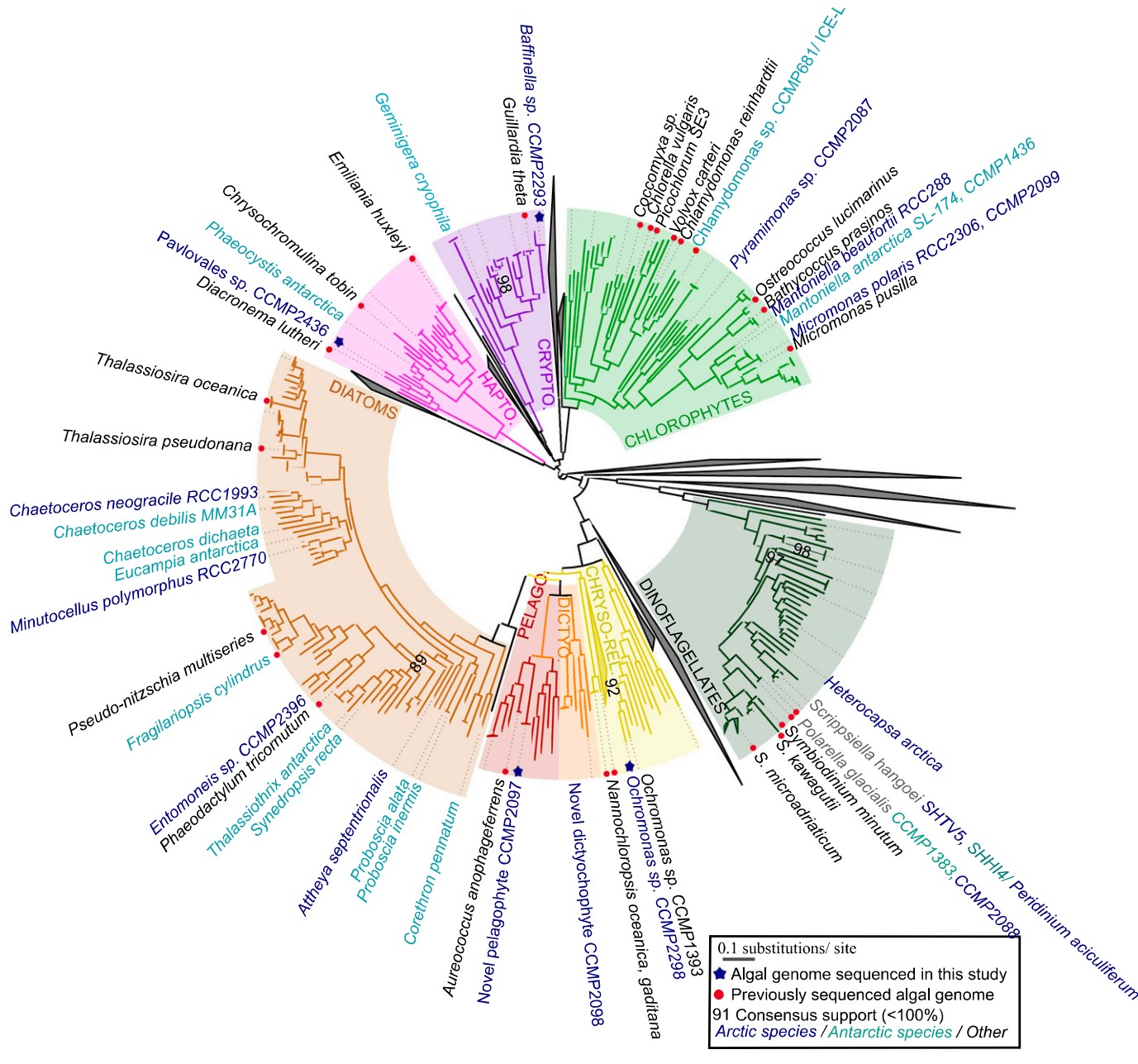

**Figure 1. Broad phylogeny of Arctic and Antarctic algae.**
Consensus ML topology of a 391 taxa × 39,504 amino acid alignment based on 250 conserved single-copy nuclear genes from available microalgal genomes and MMETSP transcriptomes. Eight algal groups (cryptomonads, chlorophytes, chrysophytes, dictyochophytes, diatoms, dinoflagellates, haptophytes, and pelagophytes) with at least one Arctic representative are included. Branch colour corresponds to the phylogeny and text colour the isolation site of each genome or transcriptome considered. Sequenced Arctic and Antarctic algal strains and taxonomically representative taxa for each algal group are labelled. Genome libraries sequenced in this study are indicted with blue asterisks.

(Supplemental Data 2, sheet 2). We furthermore noted a higher gene to PFAM ratio for each sequenced genome compared with its non-polar NRL (*Baffinella sp. CCMP2293* to *G. theta*; *Pavlovales* sp. CCMP2436 to *Emiliania huxleyi* and *Chrysochromulina tobin*; novel pelagophyte CCMP2097 to *Aureococcus anophagefferens*; *Ochromonas sp. CCMP2298* to *Nannochloropsis gaditana* and *N. oceanica*). For transcriptomes, which included a wider taxonomic range of Arctic strains, the Arctic gene count to PFAM ratio was 1.48

compared with a mean non-polar strain gene count to PFAM ratio of 1.40 Supplemental Data 2, sheet 2). Specifically, the higher gene to PFAM ratios were found for Arctic chlorophytes, dictyochophytes, and dinoflagellates compared with non-polar strains. Overall, these data suggest a greater number of genes of unknown function among the Arctic strains, as was previously noted in psychrophilic prokaryotes (42) and Antarctic eukaryotes (23, 43), consistent with this being characteristic of cold environments.

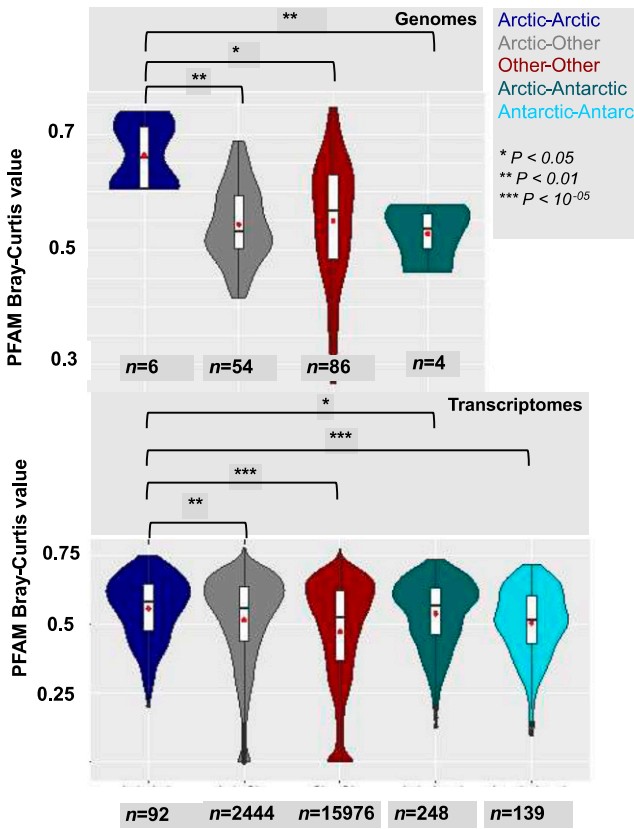

**Figure 2. Convergence of PFAM domain contents of Arctic-specific algae.**
Violin plots of Bray–Curtis indices calculated between PFAM distributions of pairs of algal genomes (top) or transcriptomes (bottom), separated by habitat: Arctic (isolation site > 60°N, within Arctic water masses), Antarctic (isolation site south of the polar front of the Antarctic Circumpolar Current), or other (all intermediate latitudes). Comparisons between members of the same taxonomic group and involving either freshwater or obligately non-photosynthetic species were excluded from the analyses. As there was only one Antarctic genome (*Fragilariopsis cylindrus*) in the pan-algal dataset, it was not considered in the genomic comparisons. Significance values of one-way ANOVA tests of the separation of means (red dots) are provided between Arctic–Arctic strain pairs and all other forms of strain pairs considered.

## PFAM within Arctic convergences

On the assumption that the geographically associated similarity of PFAMs is independent of phylogeny and indicative of convergent evolution, we tested for dissimilarity of pairs of genomes or transcriptomes within and among Arctic, Antarctic, and other categories (Supplemental Data 2, sheet 3). Bray–Curtis comparisons between species pairs within the pan-algal dataset (Supplemental Data 2, sheet 4) were carried out separately for genome-only and transcriptome-only datasets to avoid artifacts from comparing different sequence library types to one another. Additional statistical tests using BUSCO normalised data and Spearman Rank correlations (Supplemental Data 2, sheets 5–6) were done to account for biases from library completeness or extreme expansions in individual PFAMs, respectively. Convergences in the PFAM content between pairs of Arctic strains, pairs of Arctic and non-Arctic strains, and pairs of non-Arctic strains to one another were tested (Supplemental Data 2, sheet 7). The multiple pairwise comparisons of the PFAMs (Fig 2) showed that Arctic–Arctic pairs had greater

mean PFAM content similarity (genome mean 0.663; transcriptome mean 0.596) than paired Arctic and other genomes or transcriptomes (genome mean 0.541; transcriptome mean 0.556) or comparisons of other–other genome or transcriptome pairs (genome mean 0.551; transcriptome mean 0.512) (Supplemental Data 2, sheet 7). Furthermore, mean Bray–Curtis similarity between Arctic–Arctic pairs was significantly greater than that between Arctic–other pairs (one-way ANOVA, with genomes $P = 6 \times 10^{-05}$; transcriptomes $P = 0.002$) and between other–other pairs of genomes or transcriptomes (genomes $P < 0.01$; transcriptomes $P < 10^{-5}$). In addition, the pairs of Arctic strains were more similar (see above) than between pairs of Arctic–Antarctic genomes (mean = 0.527, $P = 0.0047$) or transcriptomes (mean 0.569; $P = 0.049$). BUSCO-normalised and Spearman analyses and results were coherent, showing significant within-Arctic but not bi-polar convergences (Fig S8 and Supplemental Data 2, sheet 5–7). In summary, PFAM profiles in Arctic strains were more like one another than expected through random chance, consistent with convergence in genome coding content.

As a further test of potential genomic dissimilarities, we carried out a principal component analysis (PCA; phytools R package) (44) on the genomes and the transcriptomes based on PFAM content (Supplemental Data 2, sheet 8) from the multigene phylogeny of Fig 1. PCA plots of the transcriptome results (Fig S9) are for principal components 2 and 3, as principal component 1 was strongly influenced by phylogeny. For axes 2 and 3, most major groups tended to overlap, and many of the Arctic and Antarctic algae fell within the overlap region. The close proximity of three Arctic strains, *Pavlovales* sp. CCMP2436, *Ochromonas* sp. CCMP2298, and the novel pelagophyte CCMP2097 (numbers 13, 14, and 15 in Fig S9 and Supplemental Data 2, sheet 8), suggested potential convergence in PFAM content independent of phylogeny but was not conclusive. Away from the central region, the diversity (spread) of dinoflagellates and dissimilarity of cryptomonads, haptophytes, and the dictyochophyte stood out and would be consistent with earlier divergence of these groups compared with the crown *Ochrophyta* in the dataset (26). Overall, the analysis suggests that presumed core vertically inherited genes were little altered by Arctic isolation.

### PFAM expansions, enrichments, and within-Arctic HGT

Given the significant convergence in Arctic PFAM content outside of core genes (Fig 2), we searched PFAMs that occurred significantly more frequently in Arctic compared with non-Arctic strains (i.e., Arctic-associated PFAMs; Supplemental Data 2, sheet 3). PFAMs that underwent expansions or contractions in Arctic compared with non-Arctic strains were identified via a phylogenetically calibrated analysis, Computational Analysis of gene Family Evolution (CAFE), applied to all PFAMs found in the genomes and transcriptomes across the pan-algal dataset (Supplemental Data 3, sheet 1). The frequencies of 3,858 PFAMs detected the pan-algal genomes and transcriptomes, for which expansions and contractions were found in both genome and transcriptome datasets (Supplemental Data 3, sheet 2), were then tested in Arctic and non-Arctic strains by two-tailed chi-squared tests. Seven PFAMs were significantly associated ($P < 10^{-05}$, Fig 3, first axis) and eight significantly more frequently expanded and three significantly more frequently contracted ($|\log_{10}P$ expansions $- \log_{10}P$ contractions$| > 5$, Fig 3, second axis) in Arctic strains.

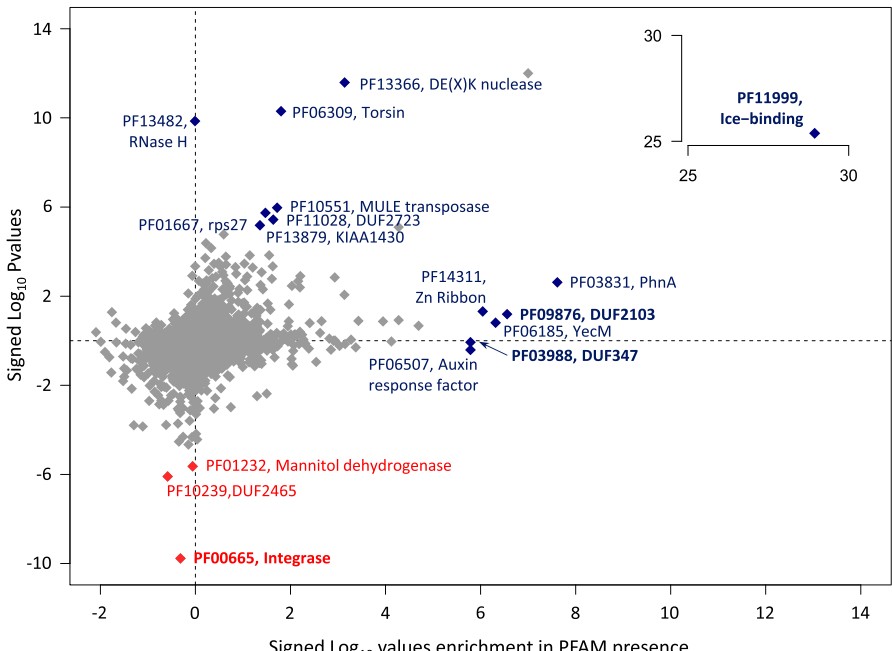

**Figure 3. Arctic-specific expansions and contractions of PFAM domains.**
Scatterplot of 3,858 PFAMs, detected in at least one algal genome and one algal transcriptome, and having inferred to have undergone at least one expansion or contraction by CAFE genome data and at least one expansion and contraction by CAFE transcriptome data, showing possible enrichments and depletions in Arctic strains. The horizontal axis shows the signed $-\log_{10}$ chi-squared $P$-value of the presence of PFAMs in Arctic versus non-Arctic species in the dataset. Positive values indicate the PFAM occurs more frequently than expected in Arctic species, and negative values indicate that the PFAM occurs less frequently than expected in Arctic species. The vertical axis shows the $-\log_{10}$ chi-squared $P$-values for enrichment in expansions of each PFAM, inferred by CAFE, in Arctic compared with non-Arctic strains, minus the $-\log_{10}$ chi-squared $P$-values of contractions in each PFAM in Arctic strains, using the same methodology. Positive values indicate that the PFAM is more frequently expanded in Arctic strains and negative values indicate that it is more contracted in Arctic strains than expected. PFAMs that are inferred to either be specifically associated (enriched in presence or expanded) or not associated (contracted) in Arctic compared with non-Arctic strains ($P < 10^{-05}$) are indicated. The insert shows PFAM (PF11999, ice-binding protein domain) which was enriched and expanded in Arctic strains and was off-scale of the main plot.

The PFAM domain PF11999 (annotated as DUF3494 and characterized as an IBP domain ([45])) was remarkable in terms of both presence (enrichment) and frequencies of expansion ($P < 10^{-25}$) (Fig 3, insert). This domain was detected in most Arctic origin algal strains in the dataset with the greatest number (100) found in the *Pavlovales* sp. CCMP2436 transcriptome (Supplemental Data 2, sheet 3), and expansions were also evident in the Arctic strains compared with related lineages of non-Arctic strains (Supplemental Data 3, sheets 1–2).

Aside from PF11999, Arctic-specific PFAMs tended to be either expanded or enriched (Fig 3). For example, PF03831 (PhnA, YjdM), which functions as an uptake protein or inducer involved in alkylphosphonate metabolism ([46], [47]), would be considered enriched as it was further to the right along the first axis. PhnA domains have previously been shown to be expanded in bacteria from the phycosphere of Antarctic seaweeds. A second example of an enriched PFAM was PF03988 (DUF347), reported as expanded in an Arctic dinoflagellate ([48]). Expanded but not enriched PFAMs include PF13482 (RNase M), whereas, for example, PF13366 DEXK nuclease was more highly expanded but also somewhat enriched. The most highly contracted PFAM in Arctic strains was PF00665—integrase, which mediates integration of viral DNA into host chromosomes and warrants further investigation along with the PFAMs containing domains with unknown functions (Fig 3).

## Phylogenies and biogeography of enriched Arctic PFAMs

To test our hypothesis that specific potentially adaptive enriched genes were acquired from other organisms in the environment, we retrieved global distributions and constructed detailed phylogenies for three previously reported polar PFAMs (PF11999, PF03831, and PF03988), which were also found to be enriched in our larger dataset (Fig 3). The three phylogenies incorporated gene models from our pan-algal dataset of 317 genomes and transcriptomes, augmented with the addition of genes from the Marine Atlas of *Tara* Ocean Unigenes (MATOU) ([49]) and UniRef ([50]), which includes bacteria and non-photosynthetic microbial eukaryotes. To our knowledge, this combination resulted in the most taxonomically exhaustive analysis of these three PFAM domains to date. The IBP domain (PF11999) was the most extensive with a 4,862-branch tree of sequences (Fig 4 and Supplemental Data 3, sheet 3). Sequences were categorized by the phylogenetic origin (Fig 4 line colour) and their documented geographic locations (Fig 4 terminal dot colour). The isoforms of the eukaryotic PF11999 sequences tended to cluster into four predominantly Arctic (clades A, B, C, and D) and two Antarctic clades (clades 1 and 2), independent of underlying phylogeny. The Arctic microbial eukaryotes did not cluster with any of the 91 Arctic-origin bacterial or archaeal IBPs in the large dataset.

Arctic clades A and B were the most phylogenetically diverse and included several IBPs from Antarctic isolates (Fig 4). The Arctic clade A included the Arctic and Antarctic *P. glacialis* CCMP2088 and CCMP1383, respectively, sequences from the "Scrippsiella-Peridinium" complex, which are now placed within a single genus *Apocalathium* ([51]). The marine form *Scrippsiella hangoei* is now *Apocalathium malmogiense*, and the freshwater form *Peridinium aciculiferum* is now *Apocalathium aciculiferum*. But to facilitate comparisons with earlier studies, we have maintained the source nomenclature. Working with a smaller dataset, Stephens et al ([24]) suggested that IBPs from the *Scrippsiella–Peridinium* complex and the two *Polarella* strains formed a distinct dinoflagellate IBP clade that would include the polar dinoflagellate *Heterocapsa arctica*. However, as the three dinoflagellate genera in our Arctic clade A are not closely related ([52]), the branching order of the IBP is inconsistent with vertical inheritance. Our new data show that the clade

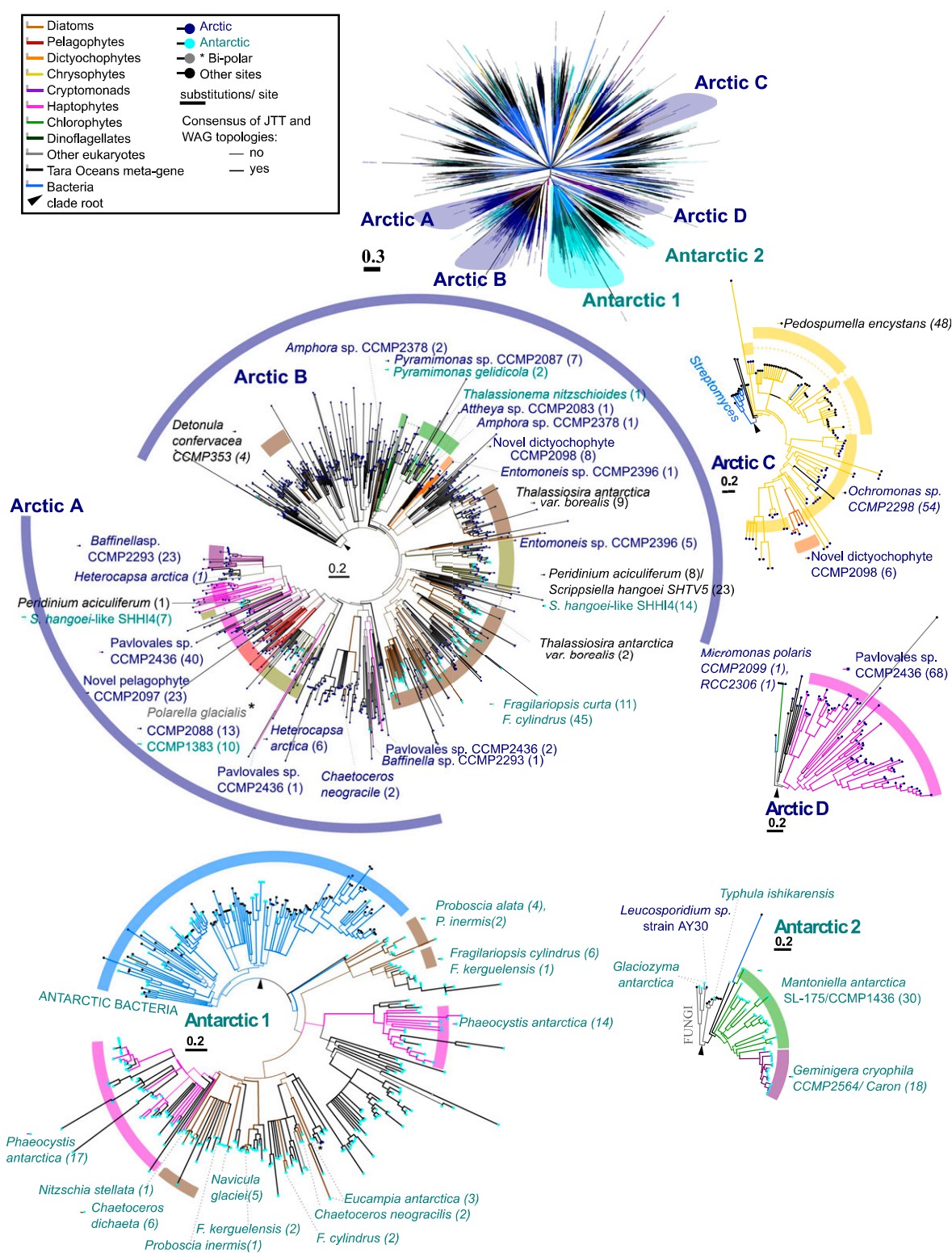

**Figure 4. HGT of ice-binding domain sequences between Arctic algae.**
Consensus best scoring tree topology obtained with RAxML under JTT and WAG substitution models for a 4,862-branch × 193 aa alignment of all ice-binding domains (PF11999) sampled from UniRef, JGI algal genomes, MMETSP, and *Tara* Oceans. Branches are shaded by evolutionary origin and leaf nodes by biogeography (either isolation location of cultured accessions where recorded or on oceanic region for which >70% total abundance of each *Tara* unigene could be recorded). One *Tara* unigene (asterisked) shows bipolar distributions (>35% total abundance in both the Arctic Ocean and the Antarctic Southern Ocean). Thick branches indicate the presence of a clade in both best scoring tree outputs. The upper tree schematic shows an overview of the global topology obtained, four clades of algal IBPs with probable within-Arctic

is dominated by Arctic environmental sequences from *Tara* Oceans and Arctic isolates, including *Pavlovales* CCMP2436, *Baffinella* CCMP2293, the pelagophyte CCMP209, and *Chaetoceros neogracile* RCC1993, which is a diatom isolated from the Beaufort Sea, consistent with at least six within Arctic eukaryote HGT events.

The second most diverse PF11999 clade (Arctic B, Fig 4) included sequences from numerous algae which were originally isolated from the North Water in 1998 (30): the chlorophyte *Pyramimonas* CCMP2087, the dictyochophyte CCMP2098, and the diatoms *Attheya* CCMP2083, *Amphora* CCMP2378, and *Entomoneis* CCMP2396. In addition, there was a sequence from an Antarctic strain of *Pyramimonas gelidicola*. The Antarctic *Pyramimonas* strains were originally isolated from ice covered coastal saline Antarctic lakes (53). Also, within the Arctic clade B was a single IBP from a Southern Ocean diatom strain of *Thalassionema nitzschioides* (54). Arctic clade B also contained multiple copies of IBP from the pan-Arctic-boreal distributed *Thalassiosira antarctica* var. *borealis* (CCMP982, isolated from 59°N) and MATOU sequences from the Arctic. The two Arctic clades A and B had a common node that included a small cluster of distinct diatom-origin IBPs from Southern Ocean strains of *F. cylindrus* and *Fragilariopsis curta* (Fig 4) and an isoform from *Thalassiosira antarctica* var. *borealis* and Arctic and Antarctic MATOUs.

The Arctic clade C (Arctic C, Fig 4) IBPs included isoforms from the Arctic dictyochophyte CCMP2098, *Ochromonas* sp. CCMP2298, and *Pedospumella encystans*, which is a cyst-forming heterotrophic chrysophyte originally isolated from an alpine lake (55). No IBPs were detected in the temperate *Ochromonas* sp. CCMP1393, which is the 18S rRNA sister of *Ochromonas* CCMP2298.

The Arctic clade D (Arctic D, Fig 4) consisted of primarily IBPs belonging to *Pavlovales* CCMP2436 that were distinct from those of the same strain within the Arctic clade A (Fig 4). The Arctic clade D IBPs further included two independent Arctic strains of the chlorophyte *Micromonas* (CCMP2099 from Baffin Bay and RCC2306 from the Beaufort Sea). Both strains are classified as *Micromonas polaris* (56), likely the most abundant picophytoplankton species in the Arctic (57). We note that these two *M. polaris* strains are the only two out of the 12 *Micromonas* strains in the pan-algal dataset with predicted IBPs (Supplemental Data 2, sheet 2), underlining the probable horizontal acquisition of IBPs in clade D between Arctic *Micromonas* and *Pavlovales*.

The Antarctic clade 1 (Antarctic 1, Fig 4) consisted of IBPs from Antarctic diatoms, especially from *F. cylindrus*, two Antarctic *Chaetoceros*, and other diatoms isolated from the Southern Ocean or coastal Antarctica. In addition, the clade included two IBPs from the Antarctic haptophyte *Phaeocystis antarctica*. A single monophyletic-origin IBP was earlier reported for *P. antarctica* (58); the presence of the second is consistent with at least one independent HGT event with Antarctic diatoms. At the root of this clade were IBPs from Antarctic bacteria (Fig 4 and Supplemental Data 3, sheet 7), as reported previously (59, 60) and interpreted as evidence

for HGT from bacteria to diatoms in the Southern Ocean. The non-phylogenetic assortment of centric and pennate diatoms may suggest infra-diatom HGT as proposed for other environmentally adaptive genes such as coding genes and iron starvation-induced proteins (61).

Although less diverse than the other clades, the Antarctic clade 2 (Antarctic 2, Fig 4) showed potential transfers between the chlorophyte *Mantoniella antarctica* and the cryptomonad *Geminigera cryophila*. *Mantoniella* is a genus in the same chlorophyte family as *Micromonas* (Fig 1). At the base of the Antarctic clade 2 were fungal (Basidiomycota)-origin sequences, consistent with a fungal origin of these algal IBPs. These included IBP from *Glaciozyma antarctica* (previously *Leucosporidium antarcticum*), which is a psychrophilic yeast isolated from Antarctic sea ice (62, 63). Closely related was the patented sequence C7F6X3 of an IBP derived from "Leucosporidium AY30" that was originally isolated from Svalbard (64). A third Basidiomycota IBP was from *Typhula ishikariensis*. *T. ishikariensis* is a snow fungus that infects plants under snow (65). The Antarctic strain is reported to be found in Antarctic ciliates, which confers protection from freezing for the ciliate (66). All three fungal strains are widely studied and used as model organisms for biotechnological applications.

The IBP topology was independently tested by a within-alignment BLAST search of the best non-same species hits to each Arctic and Antarctic algal IBP (Supplemental Data 3, sheet 4). These searches were broadly supportive, especially for the separation of Arctic and Antarctic IBPs. The 10 species that produced the greatest number of BLAST best hits to Arctic strains (>5 sequences total and having >70% best hits with Arctic or Boreal strains) were all either Arctic or Boreal origin. The only exceptions to this were some *Fragilariopsis spp. IBPs* which yielded multiple BLAST best hits against Arctic and Boreal strains (e.g., *T. Antarctica* var. *borealis*), consistent with the position of the *Fragilariopsis* IBPs that fell between the Arctic Clades A and B (Fig 4 and Supplemental Data 3, sheet 5). All the species that produced multiple BLAST best hits to Antarctic-origin strains (>5 sequences with <10% best hits to Arctic or Boreal strains) were Antarctic strains.

The retrieval of over 10 best hits between *Pavlovales* sp. CCMP2436 and *Baffinella* sp. CCMP2293, and from *Pavlovales* sp. CCMP2436 to *H. arctica* and *P. aciculiferum*, supported the grouping of Arctic clade A. Other best hits were between *Ochromonas* sp. CCMP2298 and *Pedospumella encystans*, both seen in Arctic clade C. *G. cryophila* and *M. antarctica* were positive hits with each other as seen in the Antarctic clade 2.

Across the *Tara* Ocean dataset, the relative abundances MATOUs for each of the microbial eukaryote IBP Clades in Fig 4 showed highly selective Arctic and Antarctic distributions (Fig 5). Of 1,607 polar occurrence MATOUs 1,573 with >70% of their total relative abundance in Arctic or Antarctic metagenomes (metaG) and metatranscriptomes (metaT) and only 30 occurred outside of the polar stations. Four were detected in both polar oceans (Supplemental Data 3, sheet 7). Arctic

transfer histories and two clades of algal IBPs with probable within-Antarctic IBPs (indicated as Arctic A, B, C, and D and Antarctic 1 and 2). Numbers in parentheses identify the number of non-identical branches (i.e., gene sequences) identified in each named species. The earliest diverging branch in each clade, relative to the remaining global tree topology, is marked with an arrow. From these rooting points, probable horizontal transfer events can be inferred, for example, from monophyletic groups of sequences positioned within paraphyletic groups of sequences between sister groups of species with different phylogenetic derivations.

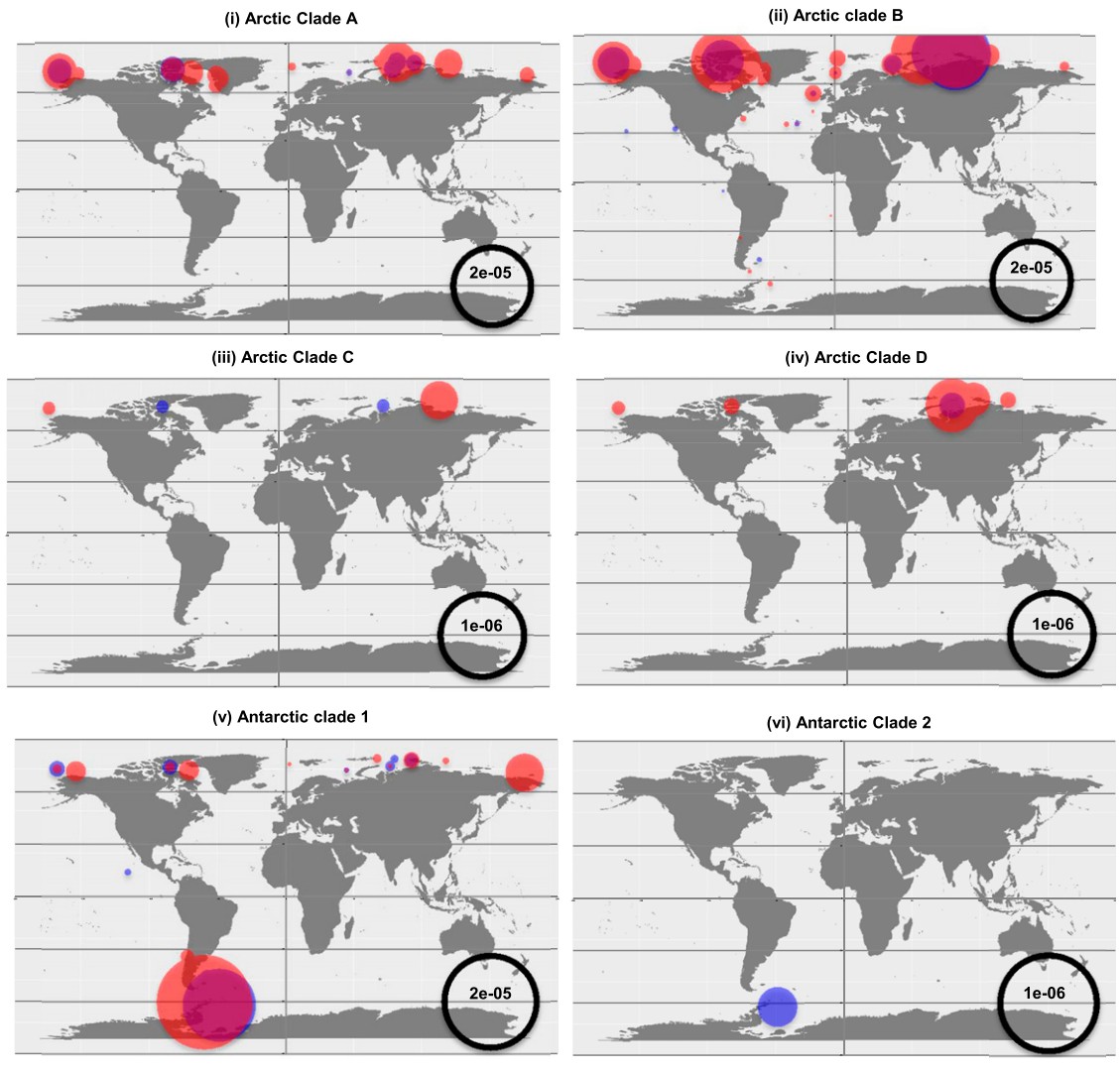

**Depth, > 0.8 μm unfiltered size fraction: Surface • DCM**

**Figure 5. Relative abundances of Tara Ocean IBP Marine Atlas of *Tara* Ocean Unigenes in Arctic and Antarctic clades.**
Marine Atlas of *Tara* Ocean Unigenes were assigned to each clade based on phylogenetic reconciliation from the consensus RAxML topology shown in Fig 4. Relative abundances, shown as a proportion of all meta-genes for each station, for surface and deep chlorophyll maximum depths based on non-size–fractionated samples (0.8–2,000 μm).

clade B MATOU IBPs were detected at low abundances in Southern Ocean stations, consistent with the Antarctic *Pyramimonas* and *Thalassionema* IBPs in that clade. The limited presence of clade B IBPs in North Atlantic stations (Figs 4 and 5, panel ii) is consistent with records for the diatoms *Detonula confervacea* and *Thalassiosira antarctica* var. *borealis* reported from the Arctic and the North Atlantic (67). The distribution and phylogeny of the IBP in these Arctic–Boreal species suggests that the same populations move between the Arctic and North Atlantic, as suggested by Luddington et al (68).

Mapping the different clades onto *Tara* Ocean data showed that the highest occurrences of IBP in the Arctic were greatest within the larger, more diverse clades A and B (Fig 5) compared with clades

C and D (Fig 5). Antarctic clades 1 and 2 MATOUs were predominantly seen in Southern Ocean stations, except for two Arctic MATOUs resolved to Antarctic clade 1 that grouped together as immediate sisters to two other MATOUs with bi-polar distributions (Fig 5).

Following from the clear IBP results, the phylogeny and biogeography of two other PFAMs with significant enrichment (Fig 3) in Arctic strains was examined. Both PhnA (PF03831) and DUF347 (PF03988) occurrences were correlated with latitude and relative abundance in the metaG and metaT data from both the Arctic and Southern Ocean *Tara* Oceans stations, suggesting functional utility of these PFAMs in polar marine waters (Supplemental Data 3, sheets 6, 8, 9). A total of 2,934 sequences (Supplemental Data 3, sheet 3) were used to construct a phylogeny of the PhnA gene (Fig S10),

which is involved in phosphonate catabolism. A polar clade stood out, consisting of mostly Arctic and Antarctic environmental sequences and included *Baffinella* sp. CCMP2293, *Pavlovales* sp. CCMP2436, *Attheya septentrionalis* CCMP2083, the novel pelagophyte CCMP2097, and the dictyochophyte CCMP2098. Several Antarctic strains, *Proboscia inermis*, *P. antarctica*, and *Chaetoceros* spp., were also within this clade (Fig S10). The small, distinct subcluster (with >66% support) consisting of CCMP2097, CCMP2098, and Arctic MATOU sequences is indicative of within-Arctic HGT. Similarly, phylogenetic inference of the DUF347 protein from a 3,942-sequence alignment (PF03988; Fig S11) revealed two microbial eukaryotic clades, each containing the bi-polar marine dinoflagellate *S. hangoei* and either Arctic (*Pavlovales* sp. CCMP2436, *Ochromonas* sp. CCMP2298, Arctic MATOUs) or Antarctic (*G. cryophila*) algal sequences, consistent with HGT events between polar algae.

# Discussion

The relative isolation and geological history of the Arctic Ocean provide an opportunity to explore geographically driven adaptive convergence in marine microalgae. The character of Arctic-specific adaptive and convergent evolution is relevant not only for understanding evolution but is critical for evaluating the fragility of the ecosystem to anthropogenically induced changes that are occurring now in the Arctic Ocean (69, 70). Our approach to unveil evolutionary trends in Arctic microalgae combined data from genomes and transcriptomes of cultured strains including four newly sequenced Arctic strains and environmental sequences from the *Tara* Oceans global survey. The concatenated multigene, 18S and 16S rRNA gene phylogenies, together with *Tara* Oceans-based reconciliation of multiple datasets indicated that the four strains chosen for sequencing were distinct from known non-Arctic species and were rare or absent from lower latitude marine environments. The Arctic prevalence highlights the unique nature of the Arctic microbiome, as suggested by earlier classic taxonomic analysis (71) and more recent omics approaches (4).

Using multiple approaches, we tested the hypothesis that Arctic microalgae convergently evolved traits or adaptations to survive in the ice-influenced ocean. In support of this, we found that, overall, the Arctic strains have comparatively large genomes and a high proportion of genes to known PFAMs compared with non-Arctic species, traits that have been previously associated with cold environments (43). Strikingly, Arctic strains in our pan-algal genomics and transcriptomics datasets had greater mean PFAM similarity compared with non-Arctic strains (Fig 2). Consistent with these observations, we document the presence of multiple Arctic-enriched and expanded PFAMs in the four new genomes and other Arctic strains (Fig 3). The highly Arctic-associated PFAM IBP (PF11999) and other PFAMs conformed to the notion that adaptive genes were acquired from other organisms in the environment. Our phylogenetically and biogeographically exhaustive search across the tree of life for homologues of genes of 3 Arctic-enriched PFAMs documented statistically credible HGT among Arctic algal strains and a distinctly different history of acquisitions among Antarctic

microbes and microalgae. This provides a previously underappreciated perspective on geographic isolation, selection and potentially adaptive radiation of key genes in an oceanic setting. An example of biologically connected ocean basin–specific HGT was the PhnA gene (Fig S10), which is involved in phosphonate catabolism and in marine systems is typically found in deep or winter waters associated with Thaumarchaeota (72). Thaumarchaeota are, however, abundant in Arctic surface water compared with other oceans (73). Also relevant to the present-day Arctic was DUF347 (PF3988, Fig S11), which is thought to be a stress-associated metal transporter (74, 75). In our analysis, this gene was overwhelmingly found in both Arctic and Antarctic *Scrippsiella* (*Apocalathium malmogiense*) but not in the freshwater strain (*P. aciculiferum* = *Apocalathium aciculiferum*), suggesting potential utility for living in more saline waters. As a metal transporter, the gene would be useful year-round in the Arctic where the concentrations of mercury and other labile metals are relatively high (76, 77). DUF347 is also tentatively associated with selenite reduction (78). The selenite cycle is relevant in the Arctic as it modulates mercury toxicity in beluga (78), which are a culturally significant food source for Inuit (79).

Relatively few of the over 100,000 described microalgal species (80) have been isolated and grow easily in culture (81), and there is accordingly a tendency to isolate the same species on multiple occasions (17). In this sense, it was not surprising that the four isolates were relatively rare in the *Tara* Oceans data, but although rare, they remain present in the Arctic Ocean environment. The novel pelagophyte CCMP2097 and *Baffinella* sp. CCMP2293, which were relatively abundant in the Arctic, also retrieved homologues from coastal boundary zone stations outside the Arctic (Fig S6). A potentially more global distribution of *Baffinella* also based on the 18S rRNA gene was previously noted, with a nearly identical 18S rRNA gene in a culture isolated from Galicia, Spain and amplicons from the South China Sea (35). We note that in the *Tara* Oceans data, these homologues were only detected using non-size–fractionated data and amplicons of the 18S rRNA gene, whereas plastid 16S ribotypes and IBP sequences for each strain had exclusively Arctic distributions (Fig 5), suggesting that the Arctic *Baffinella* sp. CCMP2293 and novel pelagophyte CCMP2097 have specific adaptations, restricting them to ice-influenced environments. Similarly, IBP from a Southern Ocean diatom strain of *T. nitzschioides* (54), which is a globally distributed morphospecies, but very common in the Arctic (67), was only found in polar oceans. The IBP distributions suggest retention of this gene in ice-associated strains but not in temperate *T. nitzschioides* or non-polar MATOUs (Fig 5). However, given that this result was based on transcriptomes, which may have not been expressed in the warmer waters, we cannot rule out that the *T. nitzschioides* MMETSP0693 (CCMP3366) isolated from a winter sample from Martha's Vineyard, MA, USA, in 1958, could express IBPs under more favorable conditions, and the history of IBP in this species remains speculative. *Pavlovales* sp. CCMP2436 and *Ochromonas* sp. CCMP2298, which were much rarer, were only found in the *Tara* Arctic stations (Fig S6). The highly diversified and Arctic-specific IBP genes detected in the two rarer strains support the notion that once acquired, the IBDomain enabled persistence of the strains in this ice-influenced region (Figs 4 and 5). The *Pavlovales* sp. CCMP2436 with 100 isoforms of IBPs suggests the utility

of diversification of this gene, which would have enabled a member of a common coastal temperate group to live in Arctic marine waters.

The IBP phylogeny provides further insight into the history and environmental functions of both Arctic and Antarctic microalgae. PF11999 remains annotated as a domain of unknown function (DUF3494) and occurs widely in bacteria and fungi in alpine and seasonally frozen habitats, where the primary role is likely to prevent within-cell ice crystal formation during winter. In freshwater environments, the IBP complex binds to water through six threonine-rich domains within a larger hydrophobic manifold and enables cells to avoid osmolysis within the cell during freezing transitions (45, 58, 82). Atmospheric transport of microbes including cryosphere fungi (62, 83) could explain the origin of the IBP gene in Antarctic clade 2 (Fig 4), with transfers of the fungal IBP to *Geminigera* and *Mantoniella* from cryophilic yeasts colonizing relatively biomass-rich sea ice (84). A source of IBP genes in the older, continent–surrounded Arctic Ocean was hinted at by the grouping of IBP genes from the alpine chrysophyte *Pedospumella lacustris* in clade C (Fig 4), suggesting a freshwater origin of IBP for clade C. More extensive analysis of related IBPs such as those detected in other chrysophytes from lakes in alpine and polar regions and snow algae such as *Chloromonas brevispina*, may eventually clarify the origin and different functions of IBP in the cryosphere (85).

The IBP expression indicated by the metatranscriptomes collected during the *Tara* Arctic cruise suggests functional utility of Arctic IBPs to prevent ice crystal formation in near-freezing and fresher ocean ecosystems. Equally, IBPs with secretory functions would be useful in sea ice species including *F. cylindrus*, facilitating algal growth in brine channels formed during freeze-up. The secreted IBPs depress the brine freezing point further and act to maintain channel structure over the polar winter (45, 85, 86). The presence of multiple IBP isoforms in algae that inhabit sea ice including *Pyramimonas* and *P. glacialis*, which are common in land-fast ice (87), may furthermore point to additional functions of IBP isoforms over seasonal transitions (ice melt and ice freeze-up). IBPs in some polar bacteria are hypothesized to be secreted and then either anchored at the outer cell surface or concentrated when cells form aggregates (88). A similar function in cyst-forming ice algae would enable the aggregated cells and cysts to rapidly sink and be deposited in the sediment during spring ice melt. Cells that are resuspended into the surface zone late in the season, facilitated by water column mixing, can start the annual cycle over once entrained into newly forming sea ice. Finally, we note that different IBP isoforms in individual species may be adapted to different cellular functions, enabling flexible life strategies. For example, in a recent salinity tolerance study of the novel pelagophyte CCMP2097, one secreted isoform was over-expressed at higher salinity (35 to 45), whereas a second isoform without secretory function was over-expressed at a salinity of 8 (31).

The IBP phylogeny moreover suggests possible migration or introduction of lineages that were first adapted to Arctic conditions and subsequently colonized Antarctic coastal regions and the Southern Ocean, where they gained more specific Southern Ocean adaptations. As an example, *F. cylindrus* likely originated in the Arctic (89), invading the Antarctic in the late Pleistocene (90). The founding *F. cylindrus* would have already gained the IBP related to those found in the Arctic clades A and B and, as previously suggested, may have acquired bacterial IBP from Southern Ocean sources (Fig 4) (91, 92). Genomic analysis of other *Fragilariopsis* spp. from both poles is needed to test such a hypothesis and explore the evolutionary history of HGT more completely in polar diatoms. The high similarity of IBP sequences in the Arctic and Antarctic *P. glacialis* strains also suggests an introduction from the Arctic towards the Antarctic. The lack of divergence of the IBP gene is surprising given the evident whole-genome divergence of the two strains (Figs 1 and 3), with the Antarctic strain having a larger genome compared with the Arctic strain (24), suggesting that the original IBP from the Arctic was optimized for living in the land fast ice habitat characteristic of *Polarella* at both poles. A third instance of introduction of Arctic strains to the Antarctic is seen in the chlorophyte *Pyramimonas* (Fig 4, Arctic B), with *Pyramimonas* CCMP2087 and the Antarctic *P. gelidicola* IBP nearly identical. *Pyramimonas* is a common ice algae, with CCMP2087 being closely related to *Pyramimonas diskoicola* which was isolated from Disko Bay, Greenland, and at the 18S level is indistinguishable from the Antarctic *P. gelidicola* (93). Finally, we note Antarctic-to-Arctic transfer predicted by MATOUs reconciled to Antarctic clade 1 (Fig 4). Although we cannot determine what species contains these sequences, the similarity suggests more recent or ongoing bi-polar exchange, possibly linked to animal migrations (94) or ship activity.

Although gene acquisition from bacteria to microbial eukaryotes seems to be frequent, enabling algae and other protists to invade new environments (95), inter-algal HGT has received less attention. As with bacteria to algal transfers, HGT among algae could occur via biotic interactions such as phagotrophic uptake by mixotrophic algae. Algae preying on other algae is common in dinoflagellates, for example (96). The four algal strains sequenced here are all predicted to be phago-mixotrophs (97), as is the Arctic *Pyramimonas* CCMP2087 (98) but other mechanisms are available and would be consistent with the presence of non-phagotrophic groups such as diatoms in the IBP polar clades. These include viral infection (99, 100) or even the direct exchange of genetic material within the Arctic water column or sea ice (58, 82). The long semi-isolation of the Arctic Ocean and the currently perennially cold conditions, with surface waters remaining around 0°C even in summer, provide an ongoing selective advantage to lineages specializing in Arctic conditions. The eukaryotic–eukaryotic transfer of genes could well be a rare event, but by providing an overwhelming advantage in the relatively closed Arctic Ocean, organisms with advantageous genes would rapidly dominate the system, as has been documented in bacteria living in extreme environments (101, 102). The patterns we see in the Arctic genomes may be exceptional, with the younger age (20 mya (103)) and higher connectivity in the Southern Ocean representing a transitional stage of HGT where bacterial gene acquisition is advantageous during initial colonization of a dynamic environment. However, eukaryote–eukaryote HGT may well manifest itself in other isolated biomes with extreme environments, such as geologically isolated cave systems (104).

The regulation and functions of individual genes inferred to have undergone within-Arctic HGT await characterization using comparative transcriptomic approaches (21, 105, 106) or through the expression and characterization of candidate genes in transformable model algae. Aside from HGT, the high gene to known

PFAM ratio (microbial dark matter (107, 108)) and relatively larger genomes of the Arctic strains suggest additional evolutionary forces in action, and it remains to be determined to what extent discrete protein functional architectures have independently and convergently evolved in the cryosphere generally and in the Arctic, Antarctic, and high-altitude regions specifically. Knowledge of the origin and diversity of genetic innovations and adaptations underpinning the capacity for small algal species to persist in ice-influenced waters is a prerequisite for gauging the environmental fragility and future ecology of this important ocean biome, where ongoing ocean warming could favor southern species invasions, especially along warming coastal sites as seasonal sea ice continues to decline (109, 110). However, winter ice cover and dark cold conditions will continue in the central Arctic for the foreseeable future and hopefully provide a refuge for the adaptive genes of the distinct Arctic pan-genome.

## Materials and Methods

### Cultures and nucleic acid isolation

The algae were all isolated from Northern Baffin Bay within the Pikialasorsuaq/North Water Polynya (29) in June 1998 using a serial selection-dilution technique until a single species was isolated. The cultures are available from NCMA Bigelow [https://ncma.bigelow.org/products/algae](https://ncma.bigelow.org/products/algae) and are cryopreserved at NCMA. For this study, we obtained the cultures from the culture collection, which were grown in L2 medium without Si in aged seawater at ca. 4°C and which is slightly above ambient surface water conditions of the latitude and longitude where they were first isolated (27) (Fig S7) and under continuous illumination of 100 $\mu$mol photons m$^{-2}$ s$^{-1}$, which corresponds to surface water light levels in June at that latitude (57). Before growing the sub-cultures used for nucleotide sequencing, the culture was transferred to medium with added antibiotics (106) to minimize bacterial contamination, which was assessed via light and epifluorescence microscopy.

Nucleic acids were harvested from the batch cultures in late exponential phase by centrifugation at 3,000$g$ for 30 min at 4°C. The supernatant was discarded, and pellets were frozen in liquid nitrogen and stored at –80°C until nucleic acid extraction. RNA and DNA were collected for whole-genome sequencing following the U.S. Department of Energy Joint Genome Institute protocols (111).

### Sequencing

Genomes in this study were sequenced on an Illumina platform using a combination of a 400–800 bp Tight Insert library with 1.5, 4, and 8 kb insert Cre-LoxP recombination libraries. For the novel pelagophyte CCMP2097, ligation-free paired-end libraries were used in addition.

For Tight Insert libraries, 2–3 $\mu$g of DNA was sheared to 400 or 800 bp using a Covaris LE220 pulse focused sonicator (Covaris) and size selected using a Pippin Prep (Sage Science). The fragments were end-repaired, A-tailed, and ligated to Illumina compatible adapters (IDT, Inc.) using a KAPA-Illumina library creation kit (KAPA Biosystems). For Cre-LoxP recombination libraries, 5–25 $\mu$g of DNA was sheared using a Covaris g-TUBE (Covaris) and gel size selected for 1.5, 4, and 8 kb, respectively. The sheared DNA was end-repaired and ligated with biotinylated adapters containing loxP, circularized via recombination by a Cre excision reaction (New England Biolabs), and randomly sheared using a Covaris LE220 (Covaris). For ligation-free paired-end libraries, 15–25 $\mu$g of DNA was sheared using HydroShear (Genomic Solutions) and gel size selected for 4.5 and 8 kb, respectively. The sheared DNA was end-repaired, ligated with biotinylated adapters, circularized by intra-molecular hybridization, then digested with T7 Exonuclease and S1 Nuclease (Invitrogen).

Sheared ligation fragments were end-repaired and A-tailed using the KAPA-Illumina library creation kit (KAPA Biosystems) followed by immobilization of mate pair fragments on streptavidin beads (Invitrogen). Illumina compatible adapters (IDT, Inc.) were ligated to the mate pair fragments and amplified with 8–12 cycles of PCR (KAPA Biosystems).

All four transcriptomes were sequenced using stranded Illumina RNA-Seq protocols. Poly(A)+ RNA was isolated from 10 $\mu$g total RNA using a Dynabeads mRNA isolation kit (Invitrogen), repeated twice to remove all residual rRNA contamination, then fragmented using RNA Fragmentation Reagents (Ambion) at 70°C for 3 min, targeting fragments around 300 bp, and purified using AMPure SPRI beads (Agencourt). Reverse transcription was performed using random hexamer primers (Fermentas) and SuperScript II (Invitrogen), with annealing, elongation, and inactivation steps of 65°C for 5 min, 42°C for 50 min, and 70°C for 10 min, respectively. Purified cDNA was used for second-strand synthesis, with a dNTP mix where dTTP was replaced by dUTP at 16°C for 2 h. Targeted (300 bp) double-stranded cDNA fragments were purified using AMPure SPRI beads, then were blunt-ended, poly A-tailed, and ligated with TruSeq adapters using Illumina DNA Sample Prep Kit (Illumina) and purified again. Second-strand cDNA was removed through dUTP digestion with AmpErase UNG (Applied Biosystems). Digested cDNA was again cleaned up with AMPure SPRI beads, amplified by 10 cycles of PCR with Illumina TruSeq primers, and finally cleaned again with AMPure SPRI beads.

Both genome and transcriptome libraries were quantified using a next-generation sequencing library qPCR kit (KAPA Biosystems) and run on a LightCycler 480 real-time PCR instrument (Roche). The quantified libraries were prepared for sequencing, using a TruSeq paired-end cluster kit (v3) and a cBot instrument (Illumina) to generate a clustered flow cell and sequenced on an Illumina HiSeq2000 sequencer using a TruSeq SBS sequencing kit, v3, after a 2 × 100 or 2 × 150 indexed run recipe.

After sequencing, the genomic fastq files were screened for phix contamination. Reads composed of > 95% simple sequences were removed. Illumina reads < 50 bp after trimming for adapter and quality (q < 20) were removed. The remaining illumina reads were assembled using AllPathsLG (112) parameters: DATA_SUBDIR = data RUN = run SUBDIR = assem1 TARGETS = standard OVERWRITE = True THREADS = 6 CLOSE_UNIPATH_GAPS = False. The resulting assembled scaffolds were screened against all bacterial proteins and organelle sequences from GenBank nr (113) and removed if found to be a contaminant. Illumina transcriptome reads were QC filtered to remove contamination and assembled into consensus sequences using Rnnotator v. 2.5.3 (114). Each genome was annotated using the JGI Annotation Pipeline, which detects and masks repeats and transposable elements, predicts genes, characterizes each

conceptually translated protein with sub-elements such as domains and signal peptides, chooses a best gene model at each locus to provide a filtered working set, clusters the filtered sets into draft gene families, ascribes functional descriptions (such as GO terms and EC numbers), and creates a JGI genome portal in PhycoCosm (https://phycocosm.jgi.doe.gov/) with tools for public access and community-driven curation of the annotation (32, 115).

### Phylogenic analyses

#### *Multigene phylogeny*

Our pan-algal dataset consisted of 21 genomes from JGI (accessed January 2018) (32), which included the four algae sequenced in this study, and 296 decontaminated MMETSP transcriptomes (33, 116). The pan-algal dataset covered the eight algal groups having at least one sequenced Arctic genome or transcriptome. These were chlorophytes (corresponding to the class Chlorophyta, containing sequenced Arctic species within the Mamiellophyceae), cryptomonads (corresponding to the phylum Cryptophyta, inclusive of *Goniomonas*), haptophytes (corresponding to the phylum Haptophyta, with sequenced Arctic members within the Pavlovophyceae), dinoflagellates (corresponding to the phylum Dinophyta), and four groups corresponding to individual or merged classes within the phylum Ochrophyta: diatoms (Bacillariophyceae), dictyochophytes (Dictyochophyceae), pelagophytes (Pelagophyceae), and "chrysophyte-related species," which included Chrysophyceae, Synurophyceae, Synchromophyceae, Eustigmatophyceae, and Pinguiophyceae. The isolation site of each sequenced strain was taken from culture collection accessions of each strain and is provided to the user via an interactive map at https://tinyurl.com/4bawubkv. The minimum and maximum growth temperatures reported for the strains were gleaned from literature searches of the cultures (Supplemental Data 1, sheet 1).

Reference sequences from published eukaryotic multigene trees (26, 34) were augmented using diamond v0.9.30.131 (117) with orthologues from all members of the pan-algal dataset. To infer phylogenies, single-gene datasets were aligned by MAFFT v7.407 (118) using the L-INS-i refinement and a maximum of 1,000 iterations, then trimmed by trimAl v1.4 (119) using the -gt 0.5 setting. ML trees were inferred by IQ-TREE v 1.6.12 (120) using the LG + F + G model. Paralogs were manually removed from single-gene datasets, considering their branching and alignment coverage. The concatenated multi-gene alignment was finally trimmed to remove sites with more than 20% gaps with trimAl (-gt 0.8), as were sites exhibiting the fastest exchange rates, which were removed using TIGER (121). Analogously to the reference multi-gene matrix (34) the final ML tree was reconstructed using the posterior mean site frequency model of IQ-TREE on an LG + F + G guide tree, with the multi-gene matrix treated as a single partition to eliminate small sample LBA bias (122). Incorporated gene and tree topologies are provided in Supplemental Data 1, sheets 2–3; and single-gene alignments and topologies, and concatenated alignments are provided in the linked supporting database https://osf.io/3pmxb/ in the directory "Supporting Data" in the folder "Multigene tree topologies."

#### *18S and 16S phylogeny*

18S rDNA and chloroplast 16S rDNA sequences corresponding to the cultured strains used for the pan-algal dataset were downloaded along with environmental sequences from the Arctic and Antarctic–Southern Ocean from GenBank nr (March 2020). These were supplemented with SS rRNA orthologues searched for each group from MMETSP nucleotide sequence libraries by BLASTn, using a randomly selected downloaded query from the algal group considered and a threshold e-value of $10^{-05}$. Homologous sequences were identified by a reciprocal BLASTn search against a complete copy of the *Arabidopsis thaliana* chloroplast, mitochondria, and nuclear genomes enriched with the query sequence; only sequences that retrieved the query as the BLAST best hit were retained (123). A complete set of query sequences is provided in Supplemental Data 1, sheet 4.

Retained homologues were aligned using MAFFT (v. 7.407) under the –auto and –adjustdirection settings and manually edited to remove non-homologous, fragmented, and chimeric sequences (118). Curated alignments were trimmed with trimAl under the -gt 0.5 and -gt 0.8 settings (119), and tree topologies were inferred from trimmed alignments using MrBayes v 3.2.1 and RAxML v 8.0 as integrated into the CIPRES web server (124, 125). MrBayes trees were run with two chains for 600,000 generations with burnin fractions of 0.5 and were manually verified in each case to have reached a final convergence statistic of ≤ 0.1 before calculation of the consensus topology, whereas RAxML trees were run by default for 450 bootstrap replicates, with automatic bootstopping function applied. Curated alignments, individual and consensus tree topologies are provided in the Supplemental Data 1, sheets 4–7, and in the linked online database at https://osf.io/3pmxb/ in the directory "Supporting Data" and the folder "18S and 16S trees."

#### *Tara Oceans analysis*

*Tara* Oceans expedition full contextual data are available at https://doi.org/10.1594/PANGAEA.875582. Ribotypes corresponding to the 18S rRNA (V4 and V9 variable regions) and 16S rRNA (V4–V5 regions) sequences of each strain were identified from version 2 of *Tara* Oceans data (126, 127) using the 18S and 16S trees defined above as a guide. Briefly, the complete 18S or 16S sequence of the species in question was searched by BLASTn against all 18S nuclear or 16S plastid sequences previously assigned to the same taxonomic group as the query (*Baffinella* sp. CCMP2293: cryptomonads; *Pavlovales* sp. CCMP2436: haptophytes; *Ochromonas* sp. CCMP2298: chrysophytes; novel pelagophyte CCMP2097: pelagophytes), with threshold e-value $10^{-05}$. Matching sequences were then searched against the complete curated 18S or 16S alignments for each group using BLASTn, and only sequences that yielded BLAST best hits against either the query sequence or its immediate sister groups in the 18S or 16S tree topology were retained for subsequent analysis.

Retained ribotypes were realigned against the reference library of cultivated 18S or 16S sequence for the corresponding algal group (Supplemental Data 1, sheet 8) using MAFFT under the –auto setting, manually trimmed to retain only the 18S V4, 18S V9, or 16S V4–V5 regions and finally trimmed with trimAl under the –gt 0.5 setting (118, 119), and trees were calculated from curated alignments using RAxML with automated bootstopping, as defined above (125). Ribotypes that were inferred from the best scoring RAxML tree topology to be more closely related to the Arctic query species than the nearest cultured non-Arctic reference and with at least 97%

nucleotide similarity to the Arctic query species assessed by BLASTn search were retained for the calculation of absolute and total relative abundances. The relative abundances are reported as the proportion of all 18S V4, 18S V9, or 16S V4–V5 ribotypes present in a sample. Curated alignments and tree topologies along with tabulated individual and total read abundances are provided in Supplemental Data 1 (sheets 10–11). Raw data pertaining to the identification of matching ribotypes by BLAST and phylogeny, and calculation of quantitative abundance trends, are provided in the linked online database https://osf.io/3pmxb/ within the directory "Supporting Data" in the folder "TARA Oceans calculations."

### Quantitative analysis of PFAM content

PFAM distributions for each algal genome in the pan-algal dataset (both newly and previously sequenced accessions) were reannotated for this study using InterProScan and an updated (December 2020) version of the PFAM database from the constituent fasta files for each genome (128, 129). PFAM annotation files for each MMETSP transcriptome were manually downloaded from the source accession and cleaned using a previously defined pipeline which compares the relative BLAST similarity between pairs of nucleotide sequences in each MMETSP library to identify transcripts of potential contaminant origin (116, 130). Tabulated PFAM outputs are provided in Supplemental Data 2 (sheet 3), and complete PFAM lists per gene for each decontaminated library are provided in the online database https://osf.io/3pmxb/ within the directory "Supporting Data" in the folder "PFAM Bray-Curtis distributions."

Phylogenetically aware PCA was performed on both genome and transcriptome datasets (44). Edited versions of the multigene tree topology (Fig 1) retaining only genome or transcriptome branches were generated using MEGA version X as phylogenetic templates (131). Output data for PCA are provided for user exploration in Supplemental Data 2, sheet 8.

Similarity in PFAM content between different algal PFAM libraries was calculated using Bray–Curtis (132, 133) and Spearman index (100, 134) approaches, and the total number of concordant PFAMs between each library pair was then normalised against the total number of complete (single copy or duplicated) eukaryotic BUSCOs retrieved in each library (135). A schematic diagram of the methodology used and the different types of convergence visible using each technique is available via a direct link https://osf.io/kh4xu/. To determine if Arctic species have converged on similar PFAM contents, pairwise Bray–Curtis indices were calculated between the PFAM distributions across the pan-algal dataset, considering genome and transcriptome libraries separately (100, 132, 133). To avoid artifacts caused by differences in phylogenetic proximity, species from the same taxonomic group were excluded before calculating mean Bray–Curtis values between pairs of libraries from different habitats (either Arctic, Antarctic/Southern Ocean, or "Other" for non-polar species).

For Bray–Curtis and Spearman calculations, algae were divided into three geographic categories: Arctic (strains isolated from > 60°N, within Arctic water masses), "Antarctic" origin strains from south of the Polar Front, defined as the 6°C sea-surface temperature isotherm, and "Other" for the non-polar sequences. To avoid introducing biases because of comparing taxa of unequal phylogenetic distance, Bray–Curtis and Spearman calculations were only

compared between pairs of algal libraries from different major taxonomic groups. The comparison was limited to marine photosynthetic species. PFAM convergence values were then used to calculate mean values and perform one-way ANOVAs of difference, within and between different groups of algae in the dataset, separated by biogeography into "Arctic," "Antarctic," and "Other" taxa, as above. Tabulated Bray–Curtis and Spearman outputs are provided in Supplemental Data 2, sheets 4–7, and all raw data, including alternative format outputs, are provided in the online database https://osf.io/3pmxb/ within the directory "Supporting Data" in the folder "PFAM Bray-Curtis distributions."

### Identification of marine Arctic-associated PFAMs

PFAMs whose presence or absence was specifically associated with Arctic strains were assessed by calculating the frequency with which the PFAM was recovered in Arctic compared with non-Arctic strains ("Antarctic" or "Other") (see Supplemental Data 2, sheet 3). These frequencies were used to calculate a ratio and a chi-squared $P$-value of enrichment of the PFAM in Arctic species in the dataset with cutoff $P$-value $10^{-05}$. Only PFAMs that were detected in both Arctic genomes and Arctic transcriptomes were considered possible candidates for enrichment. As above, the analysis focused on marine origin photosynthetic strains.

To accurately identify expansions and contractions in PFAM content across each strain, high-frequency PFAMs (defined as PFAMs for which the maximum frequency minus minimum frequency observed was greater than 100 across the entire dataset) were first fragmented into smaller orthogroups. Briefly, profiles were extracted for each high-frequency PFAM using hmmfetch (136) and then re-annotated for all libraries using hmmsearch with threshold expected value of $10^{-05}$. Proteins containing these domains were used to run OrthoFinder v 2.4.1 (137) with inflation value of 1.3 to avoid overfragmentation of orthogroups. Orthogroups were filtered (maximum – minimum frequency > 2) to remove uninformative examples; then the presence of PFAM domains (defined as presence in at least 10% of protein space within the orthogroup) was called for each orthogroup.

CAFE was performed on the composite set of low-frequency PFAMs, and orthogroups decomposed from high-frequency PFAMs, for separate sets of genome- and transcriptome-only libraries, using MEGA-edited versions of the previously generated multigene reference tree, as in the phylPCA analysis above (131). A single γ rate for the λ and error model was assessed from the low-frequency PFAM dataset across the entire phylogenetic tree. The CAFE outputs obtained were used to calculate enrichment ratios and $P$-values for "expansions," defined as signed positive CAFE scores, and "contractions," defined as signed negative CAFE scores, for Arctic strains within the dataset, using methodology as defined above. Summarised CAFE outputs and $P$-values are provided in Supplemental Data 3, sheets 1–2, and all raw data, including the prior decomposition of PFAMs into orthogroups, are provided in the online database https://osf.io/3pmxb/ within the directory "Supporting Data" in the folder "CAFE and phylPCA."

### Environmentally supported phylogenies of Arctic-associated PFAMs

The global distributions of sequences containing three PFAMs (PF03831, PF03988, and PF11999) whose presence was significantly associated with

Arctic strains in the enrichment analysis (Figs 3, 5, S10, and S11) were investigated in meta-transcriptome and meta-genome sequence data from the *Tara* Oceans, inclusive of sequences from the Arctic Ocean (49, 126). Sequences containing each PFAM were extracted using hmmer with the gathering threshold option (136, 137). These were classified into four categories based on the sum of all relative abundances (normalised against the total number of meta-T or meta-G unigenes sequenced from the sample) across all stations and size/depth fractions. These were classified as: "Arctic" sequences (>70% summed relative abundances from stations >60°N and corresponding to Arctic water masses); "Antarctic" sequences (>70% summed relative abundances contained in stations of >55°S; corresponding specifically to Tara stations south of the Polar Front); "Bipolar" sequences, (>70% summed relative abundances contained in stations of >60°N or >55°S, including >20% each in stations >60°N and stations >55°S); and "Other" (<70% summed relative abundances contained in stations of >60°N and >55°S).

The *Tara* Oceans metagenomic sequences were aligned against all sequences containing the PFAM of interest from all algal genomes and transcriptomes in the pan-algal dataset and all sequences containing the PFAM in UniRef (downloaded March 2020) (138). Alignments were carried out using MAFFT under the–auto setting followed by a more stringent round of alignment using –gap_open_penalty 12, –gap_extension_penalty 3, and –maxiteration 2 settings to remove poorly aligned sequences (118). Strain-specific sequence accessions were manually labelled with "Arctic" and "Antarctic" provenance considering the isolation site of the strain, where recorded. Alignments were manually trimmed after each step, subsequently trimmed with TrimAl using the –gt 0.5 setting (119), and finally used to infer best scoring trees with RAxML using the PROTGAMMAGTR, PROTGAMMAJTT, and PROT-GAMMAWAG substitution matrices (125). Environmental sequence calculations and individual and consensus topologies for each tree are provided in Supplemental Data 2 (sheets 3–9) and in the linked supporting database https://osf.io/3pmxb/ within the directory "Supporting Data" in the folder "Environmental PFAM calculations."

# Data Availability

The genome assemblies and annotations are available from JGI PhycoCosm portal (32) and have been deposited in DDBJ/ENA/GenBank with the following URLs and NCBI accessions: CCMP2293: https://phycocosm.jgi.doe.gov/Crypto2293_1, PRJNA223438; CCMP2436: https://phycocosm.jgi.doe.gov/Pavlov2436_1, PRJNA223446; CCMP2298: https://phycocosm.jgi.doe.gov/Ochro2298_1, PRJNA171379; CCMP2097: https://phycocosm.jgi.doe.gov/Pelago2097_1, PRJNA210205.

Collection sites for the four strains and the stations for environmental Tara data can be visualised https://www.google.com/maps/d/u/0/viewer?mid=1dLO8-_xddGCvsxPUtUAJyu00hqv3KVoJ&ll.

# Supplementary Information

# Acknowledgements

The authors thank E Virginia Armbrust (University of Washington) and Jackie Collier (Stony Brook Univ. New York) for permission to use a *Pseudo-nitzschia multiseries* genome for comparative PFAM analysis and *Aplanochytrium kerguelense*, *Aurantiochytrium limacinum*, and *Schizochytrium aggregatum* transcriptomes for phylogenetic dataset construction, respectively. All required permissions and relevant research licenses for water sampling from Canadian Coast Guard vessels were obtained from relevant national and international agencies, including associations within the Inuit Nunangat. The work proposal: 10.46936/10.25585/60001017 (C Lovejoy as principle investigator) conducted by the U.S. Department of Energy Joint Genome Institute (https://ror.org/04xm1d337), a DOE Office of Science User Facility, is supported by the Office of Science of the U.S. Department of Energy under Contract No. DE-AC02-05CH11231. C Bowler, FM Ibarbalz, and JJ Pierella Karlusich acknowledge support from ECOS Sud-Argentine program (AT08ST18). Z Füssy acknowledges support from the JW Fulbright Commission of the Slovak Republic and computational resources supplied by the project "e-Infrastruktura CZ" (e-INFRA LM2018140) provided within the program Projects of Large Research, Development and Innovations Infrastructures. C Lovejoy received support from the Discovery program of Natural Science and Engineering Council (Canada) and a pilot project grant from Genome Québec. RG Dorrell and AMG Novák Vanclová acknowledge funding from a CNRS Momentum Fellowship (2019–2021) awarded to RG Dorrell. RG Dorrell was funded by Agence Nationale de la Recherche (ANR) Young Researcher award (JCJC; ANR-21-CE02-0014 "PanArctica"). C Bowler acknowledges funding from the European Research Council under the European Union's Horizon 2020 research and innovation programme (grant agreement No. 835067; Diatomic), the French Government "Investissements d'Avenir" programmes MEMO LIFE (ANR-10-LABX-54), PSL* Research University (ANR-1253 11-IDEX-0001-02), France Génomique (ANR–10-INBS-09), OCEANOMICS (ANR-11-BTBR-0008), "the BrownCut project" (ANR-19-CE20-0020), and a Research Grant "Green Life in the Dark" (RGP0003/2016) from the Human Frontier Science Program.

## Author Contributions

RG Dorrell: data curation, software, formal analysis, supervision, investigation, visualization, methodology, and writing—original draft, review, and editing.
A Kuo: conceptualization, resources, data curation, software, formal analysis, supervision, validation, investigation, visualization, methodology, project administration, and writing—original draft, review, and editing.
Z Füssy: data curation, software, formal analysis, validation, visualization, methodology, and writing—original draft, review, and editing.
EH Richardson: conceptualization, software, formal analysis, validation, investigation, visualization, methodology, and writing—original draft, review, and editing.
A Salamov: conceptualization, resources, data curation, software, formal analysis, supervision, validation, investigation, methodology, and writing—review and editing.
N Zarevski: formal analysis, validation, investigation, visualization, methodology, and writing—review and editing.
NJ Freyria: conceptualization, data curation, formal analysis, validation, visualization, methodology, and writing—original draft, review, and editing.
FM Ibarbalz: conceptualization, data curation, software, formal analysis, validation, investigation, visualization, methodology, and writing—review and editing.
J Jenkins: conceptualization, resources, data curation, software, formal analysis, supervision, validation, investigation, methodology, project administration, and writing—review and editing.

JJ Pierella Karlusich: data curation, software, formal analysis, validation, visualization, methodology, and writing—review and editing.
A Stecca Steindorff: data curation, software, formal analysis, validation, methodology, and writing—review and editing.
RE Edgar: conceptualization, data curation, software, formal analysis, validation, and writing—review and editing.
L Handley: data curation, software, formal analysis, validation, investigation, methodology, and writing—review and editing.
K Lail: data curation, software, formal analysis, validation, investigation, methodology, and writing—review and editing.
A Lipzen: data curation, software, formal analysis, validation, methodology, and writing—review and editing.
V Lombard: data curation, software, formal analysis, validation, investigation, methodology, and writing—review and editing.
J McFarlane: data curation, validation, methodology, and writing—review and editing.
C Nef: data curation, formal analysis, validation, methodology, and writing—review and editing.
AMG Novák Vanclová: data curation, software, formal analysis, validation, methodology, and writing—review and editing.
Y Peng: data curation, software, formal analysis, validation, and writing—review and editing.
C Plott: data curation, software, formal analysis, validation, methodology, and writing—review and editing.
M Potvin: data curation, validation, investigation, methodology, and writing—review and editing.
FRJ Vieira: data curation, software, formal analysis, validation, methodology, and writing—review and editing.
K Barry: resources, data curation, formal analysis, validation, methodology, project administration, and writing—review and editing.
C de Vargas: conceptualization, resources, data curation, supervision, funding acquisition, validation, methodology, project administration, and writing—review and editing.
B Henrissat: data curation, formal analysis, supervision, validation, and writing—review and editing.
E Pelletier: resources, data curation, software, formal analysis, supervision, validation, investigation, methodology, and writing—review and editing.
J Schmutz: resources, data curation, formal analysis, supervision, funding acquisition, validation, investigation, methodology, project administration, and writing—review and editing.
P Wincker: resources, data curation, supervision, funding acquisition, validation, investigation, methodology, project administration, and writing—review and editing.
JB Dacks: conceptualization, formal analysis, supervision, validation, investigation, methodology, and writing—original draft, review, and editing.
C Bowler: resources, supervision, funding acquisition, validation, investigation, project administration, and writing—original draft, review, and editing.
IV Grigoriev: conceptualization, resources, data curation, software, formal analysis, supervision, funding acquisition, validation, investigation, methodology, project administration, and writing—original draft, review, and editing.
C Lovejoy: conceptualization, resources, data curation, software, formal analysis, supervision, funding acquisition, validation, investigation, visualization, methodology, project administration, and writing—original draft, review, and editing.

## Conflict of Interest Statement

The authors declare that they have no conflict of interest.

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
