## [Reviewer comments · Life Science Alliance]

Life Science Alliance

Convergent evolution and horizontal gene transfer in Arctic Ocean microalgae

Richard Dorrell, Alan Kuo, Zoltan Fussy, Elisabeth Richardson, Asaf Salamov, Nikola Zarevski, Nastasia Freyria, Federico Ibarbalz, Jerry Jenkins, Juan Pierella Karlusich, Andrei Stecca Steindorff, Robyn Edgar, Lori Handley, Kathleen Lail, Anna Lipzen, Vincent Lombard, John McFarlane, Charlotte Nef, Anna Novak Vanclova, Yi Peng, Chris Plott, Marianne Potvin, FABIO Vieira, Kerrie Barry, Colomban de Vargas, Bernard Henrissat, Eric Pelletier, Jeremy Schmutz, Patrick Wincker, Joel Dacks, Chris Bowler, Igor Grigoriev, and Connie Lovejoy

DOI: <https://doi.org/10.26508/lsa.202201833>

Corresponding author(s): Connie Lovejoy, Université Laval

Review Timeline:	Submission Date:	2022-11-16
	Editorial Decision:	2022-11-17
	Revision Received:	2022-11-29
	Accepted:	2022-12-01

Scientific Editor: Novella Guidi

Transaction Report:

Please note that the manuscript was previously reviewed at another journal and the reports were taken into account in the decision-making process at Life Science Alliance.

Reviewer #1 (Evidence, reproducibility and clarity (Required)):

The authors have assembled an enormous amount of statistical data on the genomes and phylogeny of Arctic algae, including the genomes of four new species that they sequenced for this study. Their main finding is that horizontal gene transfer has led to convergent evolution in distantly related microalgae.

****Major comments****

Reviewer #1: The purpose of the study is not clearly stated in the abstract or the introduction. The authors say (line 93) "Defining the genetic adaptations underpinning these small algal species is crucial as a baseline to understand their response to anthropogenic global change (Notz & Stroeve,2016)." Is this their goal? Or are they just quoting another study? The authors state (line 103) "We extend by sequencing the genomes of four distantly related microalgae...". This is not really a question or a hypothesis. I am sure the authors can provide a more compelling reason to embark on such a labor-intensive study.

Reply: We agree that the aim was lost in the details and the Introduction is now focused towards the original goal of the study, which was to investigate convergent evolution in a biogeographically isolated ocean. Additional references on the formation and history of the Arctic Basin have been added to the Introduction to provide context. "An ocean has been present at the pole since the beginning of the Cretaceous. Shaped by tectonic processes (Nikishin *et al.*, 2021) the Arctic Ocean has been a relatively closed basin since the Masstrichtian at the end of the late Cretaceous epoch (ca. 70 million years before present), with episodic sea-ice cover since that time (Niezgodzki *et al.*, 2019). This long history suggests limited gene flow from the global ocean over vast time scales and Arctic marine species including microalgae could well have unique adaptations to cold arctic conditions." Line 78-83.

And following this we provide a clear hypothesis "The potential for lineages of ancient Arctic origin and the episodic input of outside species led us to our hypothesis that Arctic microalgae convergently evolved traits or adaptations aiding survival in an ice-influenced ocean. Line 112-117.

We also discuss both the adaptive and distinct physical environment of the Arctic, and its topographical separation from other ocean regions as dispersal limitation would enhance the Arctic-specific genomic signatures. We now cite the recent paper by Sommeria-Kline *et al.* (2020), which puts eukaryotic plankton biogeography into a global context (Line 72)

Reveiwier #1: The most prominent shared trait that the authors found are genes for

ice-binding proteins. However, in view of their importance, little information is given about their different types and possible functions.

Reply: We appreciate the comment and have added information on relevant ice binding proteins found in the Arctic Algae. In addition, we discuss how the functional and secretory diversity of IBP would enhance the survivability of pelagic taxa. Lines 534 to 564.

Although ice binding proteins from multicellular animals and plants are outside the scope of this study, there is a recent review; Bar Doley, Braslavsky and Davies 2016 Annual review of Biochemistry 85: 515-542.

Reviewer #1: The HGT of ice-binding proteins is a major focus of this study, but little is said about what previous studies have said about this. What are the previous studies, what are their findings and how do the present findings contribute to this?

Reply: We agree that this aspect should have been more visible. We incorporated new data to characterize IBPs drawn from MMETSP transcriptomes, and environmental Tara Ocean metagenomes, as well as our Arctic strains. We note that as we take a PFAM-based approach, the IBPs treated are DUF3494/PF11999 domain, which are type 1 IBPs / algal IBPs (Raymond and Remia 2019). As an example of novelty, we identify the position of IBPs from dinoflagellates, within a larger Arctic Clade that included CCMP2293, CCMP2436 and CCMP2097 and Arctic TARA IBP, rendering this a pan-algal IBD clade.

In addition, we were able to resolve the position of anomalous *F. cylindrus* IBP that fell between two Arctic associated clades (A and B, in our Fig 4). This finding is consistent with *F. cylindrus* originating in the Arctic as previously suggested and subsequently invading the Southern Ocean. The recurrent acquisition of multiple diverse IBP isoforms in individual species through HGT events has not been previously reported, and the extent of isoforms in the Arctic was surprising. See for example multiple different IBP forms with separate origins in Pavlovales CCMP2436 (Fig 4). The previous studies are referred to in the context of the phylogeny of the IBD within the results section: Lines 322- 413, and Lines 534-585.

Reviewer #1: Figure 5 on HGT of ice-binding proteins is difficult to follow. It would be clearer if each panel could be described separately, clearly stating its main finding. I doubt that a reader could look at this figure and explain to a colleague what it shows.

Reply: We have revised/rearranged the figure (now Fig 4) with Arctic A, B, C and D clearly indicated as well as the two Antarctic dominated clades. The upper schematic includes the deepest phylogeny of algal IBDs to date, incorporating all of UniRef, MMETSP and TARA Oceans. The fasta files underlying the tree and the nexus file used are provided in the S1 Data Folder, which is an excel folder with information on the analysis of the data. The callout and order of the clades has been revised to facilitate interpretation of the phylogenies more clearly. The entire section has been completely rewritten.

Reviewer #1: This is also a problem with many of the other figures. For each figure, what is the question being asked and what is its take-home message?

Reply: We agree that the message was lost and have now focused on our original question in our accepted proposal to JGI. "Is there a convergence among arctic microalgae at the genomic level?". We found some genome properties were common among the Arctic isolates (more unknown PFAMS and several expanded PFAMs). The importance of ice binding proteins in Arctic Isolates and the widespread inter-algal HGT of this important protein among the Arctic strains. The IBP biogeography and phylogeny strongly indicate that the Arctic microalga have acquired IBP locally and that the Antarctic strains have acquired additional isoforms independently from Antarctic bacteria and fungi (Lines 565-585).

Reviewer #1: The paper has more data than a reader can absorb. It could be strengthened by reducing the number of figures, simplifying them if possible, and more clearly stating the value of the remaining figures.

Reply. As suggested, we have refocused the paper, removing more speculative statistics based analysis and associated figures. The main conclusions are supported by the 5 main figures. We are now present 5 main figures and 11 supplementary figures (previously 23 downloadable supplementary figures and 40 on-line only figures supporting the support figures). We agree with the reviewer, and we feel the revised version is a more transparent synthesis. Briefly the Figures illustrate the following points. Fig. 1. The multigene tree of available algal genomes and transcriptomes provides a clear framework for judging the divergence of subsequent individual gene and PFAMs phylogenies. Fig. 2 (originally Fig. 3). Indicates the convergence of PFAM domains in the Arctic strains, in contrast to strains from elsewhere. Fig. 3 (originally Figure 4) shows Arctic specific expansions and contraction of PFAM domains, again demonstrating convergent evolution in the Arctic. The figure identifies specific PFAMs that contribute to the within-Arctic convergence. This figure is based on statistical methods independent of Fig 2. Figure 4 is the most extensive IBP phylogeny to date and has been discussed above. Figure 5, which was supplementary in our non-peer reviewed version, shows the biogeographic distribution of IBP, and can be compared to the distributions of the 18S rRNA genes from the four Arctic algae provided as supplementary (S6 Fig.)

****Minor comments**Reviewer #1**

1. The figure citations are confusing. E.g., what does "Fig.1- Figure supplement 1" refer to? Does this refer to 1 or 2 figures? Apparently, it refers only to Fig. S1, so many readers will be confused when they look at Fig. 1.

Reply: We apologize for the confusing format; the manuscript had been formatted for the online journal eLife. Our revision follows the more traditional style of PLoS Biology and other Review Commons journals.

2. Multiple citations should be in order of publication date, not alphabetical order.

Reply ; We agree that date of publications is quite standard and recognizes priority of publication. Several on line journals no longer follow this rule and citation order will follow the specific style used by our accepting journal.

Reviewer #1 (Significance (Required)): It is well known that useful genes tend to be shared among microorganisms. The present study strengthens previous studies in showing that gene transfer is an important process in polar regions.

Reply: We thank the reviewer for recognizing the importance of our study.

Reviewer #2 (Evidence, reproducibility, and clarity (Required)):

This manuscript is the result of a large international collaborative effort, including the US Department of Energy Joint Genome Institute. Its focus is comparative genomics of eukaryotic Arctic algae. The primary data described in the ms are four new genome and transcriptome sequences from diverse Arctic algae, represented by a cryptomonad, a haptophyte, a chrysophyte, and a pelagophyte.

The authors compare these new data to previously published genomic/transcriptomic data from eukaryotic algae with the goal of understanding genome evolution in the Arctic. The results of the paper are a series large-scale comparative genomic bioinformatics analyses, including the associated statistical analyses. The key findings center on statistically significant features of Arctic genomes, features that stand out as compared to the genomes of algae that are not primarily found in the Arctic. Together, these findings allow the authors to make various hypotheses and suggestions about genetic adaptations to polar environments.

By far the most significant finding is that the genomes of Arctic algae are enriched in genes encoding proteins with an ice-binding domain, paralleling findings from Antarctic algae. These genes appear to have spread among Arctic algal genomes via horizontal gene transfer, which raises a series of interesting questions. In my opinion, the major conclusions of this paper are supported by the data. Listed below are a few comments that may improve the ms:

Reviewer #2.

1) In today's post-genomics era, everyone seems to be sequencing nuclear genomes. Often what distinguishes high-impact and low-impact genome papers is the number of genomes presented and the quality of the genome assembly. I may have missed it, but reading the main text, the figures/tables, and the supplementary data I was not able to get a sense of the quality of the four genome assemblies from

which the main findings are based. I was eventually able to find this information from PhycoCosm (note: some of the links to this site are not working in the ms). My quick scan of the PhycoCosm summary info for the four genomes indicates that the assemblies are highly fragmented, likely because they are based on short-read Illumina sequencing rather than a combination of short and long reads. I think it is important to briefly discuss (and or present) the quality of the assemblies in the ms and to highlight the potential limitations/drawbacks of employing highly fragmented assemblies when carrying out large-scale comparative genomics.

Reply: We agree and the data concerning the genome quality assemblies has been moved to the main text Table 1. The comparison with other paired related strains is provided in an excel folder designated S2 Data Folder.

Reviewer #2.

2) Horizontal gene transfer is undeniably a major driving force in evolution, and one that has shaped genomic architecture across the Tree of Life. I believe the data presented here support a role for HGT in the genome of evolution of Arctic algae, particularly with respect to genes encoding proteins with an ice-binding domain. However, we can all think of numerous instances when authors of genome papers were too quick to point to HGT. Thus, I would urge more caution and balance when presenting the HGT data, including some discussion about factors that could incorrectly lead researchers to conclude a significant role for HGT, such as contamination, gene duplication, mis-assemblies, etc. I'm not suggesting that you change the main conclusions, but just tone down the language in places (e.g., "we reveal remarkable convergence in the coding content ...").

Reply: We understand the reviewers concerns and now more clearly outline the pipeline we have used to identify HGTs. This included: filtering each genome to remove all possible contaminant sequences first, considering both contig co-presence of vertical- and horizontally-derived genes, and reciprocal and independent annotations of gene sequences in both genome sequences and MMETSP transcriptomes. Retained genes were subjected to simultaneous BLAST analysis and manually curated phylogenies using decontaminated reference datasets. The most parsimonious explanation for our final IBP domain microbial algal clusters (Fig 4) is HGT. On the side of caution, we removed the entire section that identified potential arctic HGT based primarily on a less targeted broad statistical analysis. The focus is now on 3 genes that have clearly identifiable utility in the Arctic, were found to be enriched in Arctic genomes via a separate analysis, and had homologs in the Tara Ocean Polar circle data. In addition, we describe more clearly the role of expansion and enrichment of PFAMs and the high proportion genes without an identifiable PFAMs in the Arctic strains as evidence for arctic convergence separate from potential HGT.

Reviewer #2.

3) The downside of studying protists (as compared to multicellular animals, for instance) is that most are not widely known by the scientific community and even

fewer scientists can picture what they actually look like (e.g., Pavlova sp. CCMP2436). A few more details about the four Arctic algae that make up the focus of this paper might be helpful for the casual reader. My sense is that if at the next departmental meeting I asked my colleagues what a pelagophyte was most would look at me with a blank stare. Moreover, am I right to assume that all four algae are psychrotolerant rather than psychrophilic (Supplement Fig. 1 makes me think otherwise). It might be good to point out the difference in the text.

Reply: High resolution images of each strain are available on the JGI home page for each alga, given the multiple figures we feel photos would not add information.

Reviewer #2

4) I don't think Supp. Table 1 (the Pan-algal dataset) got uploaded correctly during the manuscript submission stage. The first link I click on gives me Supp. Table 2.

Reply, We apologize for this, the format was incorrect for the file designation and there were lost links. We now more actually refer to these as Data Folders as they are excel folders containing multiple sheets, All supplementary links will be verified again on final submission.

Reviewer #2 (Significance (Required)):

By far the most significant finding from this paper is that the genomes of Arctic algae are enriched in genes encoding proteins with an ice-binding domain, paralleling findings from Antarctic algae. These genes appear to have spread among Arctic algal genomes via horizontal gene transfer, which raises a series of interesting questions. This is not the first paper to present these types of ideas, but it is arguably the broadest analysis yet, at least with respect to eukaryotic algae. This work will be of great interest to polar scientists, phycologists, protistologists, and the genomics community. I am genome scientist studying protists, including algae.

Reply. We thank the reviewer for their insightful comments.

Reviewer #3 (Evidence, reproducibility and clarity (Required)):

****Summary:****

This manuscript is focused on Arctic microalgae, an important yet understudied community in permanently cold ecosystems. By sequencing the genomes of four phylogenetically diverse and uncharacterized polar algae, the authors seek to elucidate genomic features and protein families that are similar in polar species (and differ from their relatives from temperate environments) This work used high-throughput genomic sequencing and computational analysis to demonstrate significant horizontal gene transfer (HGT) in several gene families, including ice-binding proteins. The authors suggest that this HGT is an effector of environmental adaptation to Arctic environments.

****Major comments and experiment suggestions:****

The authors conclude that HGT between arctic species is a driver of polar adaptation. The authors strongly support the claim that HGT is present more frequently in the polar algae examined here. Whether this is adaptive should be further explored though. For instance, ice-binding domains were one PFAM group found at significantly higher frequencies in the polar species - but are all of these species associated with ice? What would be the benefit of IBDs in an alga that is found in the open ocean. Similar with the other domains (Lns 333-335), its not clear whether these are truly adaptive features. This is more speculative.

Reply: We agree that some detail was lacking and have considerably expanded our introduction on the character of the Arctic Ocean and have stated the goals and underlying hypothesis. Briefly, all surface water organisms that live in the Arctic encounter ice during the year as the ocean freezes in winter, and surface waters remain around negative 1.7 °C for much of the year. This information has been added to the introduction. We have also expanded the discussion on the multiple effects of different IBPs that would be ecologically beneficial for plankton as well as ice-algae and cite relevant experimental studies and reviews.

Reviewer #3) HGT was a major conclusion of this study, putting this in a wider perspective would strengthen the conclusion, especially in the context of HGT from prokaryotes. Are there insights on whether IBDs are present in Arctic prokaryotes?

Reply: This is a good question, and we now point out that there were 91 Arctic bacterial and archaeal IBP sequences in our comparative dataset. In contrast to the Antarctic clades, none were closely related to the Arctic strain IBPs (Fig 4). Line 336.

Reviewer #3) The data obtained from the genomic works supports the conclusions stronger than ones from transcriptomes, where what genes/domains are present would depend largely on the sampling conditions. This should be emphasized.

Reply: The main rationale for using transcriptomes was that more of these are available and enabled us to detect convergences and HGT across a broader taxonomic range, than would be possible with genome-only data, where we had access to a total of only 21 microalgal genomes. In general transcriptome studies are aimed at identifying responses under different conditions and rely on comparative expression data, usually 2 fold differences in up or down expression under different growth conditions, see for example Freyria et al. 2022 (Communications Biology). Unlike a transcriptome expression study, our data mining detected any constitutive expression in these unicellular haploid cells, we would have detected genes used under any condition that an alga happened to be growing. IBD was not detected in any of the temperate genomes, and only detected in transcriptomes of Arctic and Arctic-Boreal groups. However we agree that there may be a limitation of transcriptomes only studies. Lines 522-528.

Reviewer #3) An experiment to determine whether the species are cold extremophiles (psychrophiles) would be useful here to strongly support the data in Figure 1. The authors state that their species can not survive >6C but this is based on experiments done on older studies. Considering the cultures have been maintained as a continuous culture for decades, confirming that they still have psychrophilic characteristic would be useful. This is a straightforward and low cost experiment that requires simply measuring growth rates at several temperatures to define the optimal and confirm that the cells are not viable above 6C.

Reply: These are interesting points, and the broad “background” statements in the original manuscript would require a separate study and have been deleted. Temperature tolerance experiments are not so simple for cold adapted algae with slow growth rates. Such experiments require specialized incubators to maintain low temperatures. Temperature experiments have been carried out on the cultures in the context of other studies, see for example, Daugberg et al. 2018, J. Phycol. But this is not within the scope of the present study.

We now restrict our conclusions to the specific question of convergence among Arctic strains. We apologize for the misunderstanding on the history of the cultures. They have not been in “continuous culture” but are cryopreserved. We now simply indicate that they grow below 6 °C, which is sufficient to assume that they likely cryophiles, our experience is that they do not grow well or at all at higher temperatures, our efforts have been to maintain the cultures that are otherwise easily lost. We now make no claims about optimality or limits. Here we simply examined genomes and available transcriptomes that were generated from algae growing at 4-6 °C.

Reviewer #3) **Minor comments:**

Defining the species used here as psychrophiles would put the study in context better. The authors relate their finding to Antarctic species (HGT, ice-binding domains, large genomes) all of which are confirmed psychrophiles.

Reply: The temperature definition of psychrophiles is surprisingly high (optimal growth below 15 °C) and this definition of psychrophiles is now given in the introduction. The point is really that there are few isolates from cold surface waters that have been well studied. We now add. “A handful of polar algal genomes have been extensively studied, with 4 of these from around Antarctica and classified as psychrophiles (not being able to grow above 15 °C (Feller & Gerday, 2003)” . Lines 103-107.

Reviewer #3) A short rationale on why these species at all would be useful - are they representative of their classes? Do they have psychrophilic characteristics that might make them useful models in the future? Are they widely used now?

Reply: We appreciate the point as the definition of utility in discovery-based science is an open dialog. We agree that the study requires context and have added our rationale for selecting the species for genome sequencing to the introduction. “To address questions on genetic

adaptations to this ice-influenced environment, we sequenced 4 phylogenetically divergent microalgae, from 4 algal classes belonging to 3 algal phyla: Cryptophyceae (Cryptophyta), Pavlovophyceae (Haptophyta), Chrysophyceae and Pelagophyceae (both in the Ochrophyta) isolated from the ca. 77 °N, where surface ice flow persists through June (Mei *et al.*, 2002). The four isolates were selected as representatives of different water and ice conditions and phylogeny from available strains collected in April and June 1998 during the North Water Polynya study”.

Reviewer #3) Starting algal cultures were maintained in a continuous culture since 1998 and under continuous light since at least 2015, have the authors confirmed that these algae retain their physiological features even after this long time? The accumulation of mutations is a possibility here.

Reply: We apologize for the misunderstanding of the timeline; the history of the cultures was not given in the manuscript and the inferred history is not quite correct. The 2015 date was the year of publication for the MMETSP data. Our continuous light statement is a record of our standard culture conditions. We now elaborate on the material used in the current study. The cultures were deposited in the Bigelow culture collection (now NCMA) in 2002 and cryopreserved once they had been verified and given a culture designation. We obtained fresh cultures in 2005 and these were used for the MMETSP project. We obtained fresh cultures again in 2011, specifically for the JGI genome project. These algae do not grow fast and most of the DNA was sent to JGI in 2012 for most of the isolates. This history is rather long and not relevant, since one would speculate that over the years the algae would tend to lose the ice associated functionality, e.g. they were not frozen in seawater every year for 4 to 6 months or subject to sudden freshwater exposure, when ice melts. We would encourage other researchers to order the cultures and run experiments. We note that many of the 40 or so algae isolated from the same campaign have been used by others for specific studies and at least 8 are in the MMETSP data set. The presence of 18S rRNA and phylogenetic position of the IBP sequences compared to Tara Arctic circle data confirms long-term Arctic presence of each species and the IBP domains in the Arctic without marked changes over the last 20 years.

Reviewer #3) Ln381 - The culture collection IDs for each sequenced species should be included here

Reply: we have added the culture IDs throughout.

Reviewer #3) Ln. 389 - Algal cells are harvested and used for nucleic acid extraction, the nucleic acids themselves are not harvested

Reply: we agree and corrected the wording

Reviewer #3 (Significance (Required)):

This study is well places in the current state of research on polar alga and represents

a significant and very valuable addition to the current knowledge pool. Algae in general are lagging behind other groups of photosynthetic organisms in the number of sequenced and analyzed genomes, despite algae being one of the main primary producers globally. This is even more strongly felt in polar research, where only 4 species have been sequenced, most of which are restricted to Antarctica. There is a true gap in our knowledge when it comes to Arctic species, and this study fills this gap. As the authors correctly state, we need more knowledge on polar environments and the primary producers that support these important ecosystems in light of current climate change trends.

Reply: we appreciate the succinct summary of our study and thank the reviewer for insights and suggestions that have improved the manuscript.

Reviewer field of expertise: Polar algae, stress responses, plant and algal energetics, cell signalling

References

- Daugbjerg N, Norlin A, Lovejoy C. *Baffinella frigidus* gen. et sp. nov. (*Baffinellaceae* fam. nov., Cryptophyceae) from Baffin Bay: Morphology, pigment profile, phylogeny, and growth rate response to three abiotic factors. *Journal of Phycology*. 2018;54(5):665-80
- Feller, G. and Gerday, C. (2003) Psychrophilic enzymes: Hot topics in cold adaptation. *Nat Rev Microbiol*, **1**, 200-208.
- Freyria NJ, Kuo A, Chovatia M, Johnson J, Lipzen A, Barry KW, et al. Salinity tolerance mechanisms of an Arctic Pelagophyte using comparative transcriptomic and gene expression analysis. *Communications Biology*. 2022;5(1). doi: 10.1038/s42003-022-03461-2
- Mei, Z. P., Legendre, L., Gratton, Y., Tremblay, J. E., Leblanc, B., Mundy, C. J., Klein, B., Gosselin, M., Larouche, P., Papakyriakou, T. N., Lovejoy, C. and Von Quillfeldt, C. H. (2002) Physical control of spring-summer phytoplankton dynamics in the North Water, April-July 1998. *Deep-Sea Research Part I-Topical Studies in Oceanography*, **49**, 4959-4982.
- Niezgodzki, I., Tyszka, J., Knorr, G. and Lohmann, G. (2019) Was the Arctic Ocean ice free during the latest Cretaceous? The role of CO₂ and gateway configurations. *Global and Planetary Change*, **177**, 201-212.
- Nikishin, A. M., Petrov, E. I., Cloetingh, S., Freiman, S. I., Malyshev, N. A., Morozov, A. F., Posamentier, H. W., Verzhbitsky, V. E., Zhukov, N. N. and Startseva, K. (2021) Arctic Ocean Mega Project: Paper 3-Mesozoic to Cenozoic geological evolution. *Earth-Science Reviews*, **217**.

November 17, 2022

RE: Life Science Alliance Manuscript #LSA-2022-01833-T

Connie Lovejoy
Université Laval
Département de Biologie
Québec, QC G1V 0A6
Canada

Dear Dr. Lovejoy,

Thank you for submitting your revised manuscript entitled "Convergent evolution and horizontal gene transfer in Arctic Ocean microalgae". We would be happy to publish your paper in Life Science Alliance pending final revisions necessary to meet our formatting guidelines.

- please upload your supplemental figures as single files
- please add a summary blurb and a category to our system
- please add the Twitter handle of your host institute/organization as well as your own or/and one of the authors in our system
- please add a separate conflict of interest statement to your main manuscript text
- please use the [10 author names, et al.] format in your references (i.e. limit the author names to the first 10)
- please rename the section Data Deposition with Data Availability

A. FINAL FILES:

B. MANUSCRIPT ORGANIZATION AND FORMATTING:

Sincerely,

December 1, 2022

RE: Life Science Alliance Manuscript #LSA-2022-01833-TR

Prof. Connie Lovejoy
Université Laval
Département de Biologie
1045 ave de la médecine
Pavillon Vachon
Québec, QC G1V 0A6
Canada

Dear Dr. Lovejoy,

Thank you for submitting your Research Article entitled "Convergent evolution and horizontal gene transfer in Arctic Ocean microalgae". It is a pleasure to let you know that your manuscript is now accepted for publication in Life Science Alliance. Congratulations on this interesting work.

DISTRIBUTION OF MATERIALS:

Again, congratulations on a very nice paper. I hope you found the review process to be constructive and are pleased with how the manuscript was handled editorially. We look forward to future exciting submissions from your lab.

Sincerely,
